# PARSE-Ego4D: Personal Action Recommendation Suggestions for Egocentric Videos

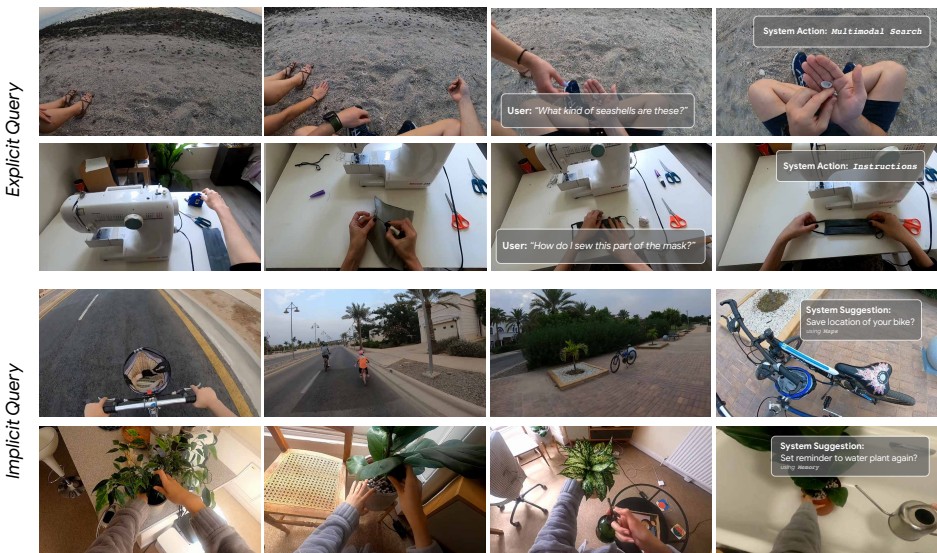

Figure 1: Examples of action suggestions for different videos in the PARSE-Ego4D dataset.

## Abstract

Intelligent assistance involves not only understanding but also action. Existing ego-centric video datasets contain rich annotations of the videos, but not of actions that an intelligent assistant could perform in the moment. To address this gap, we release **PARSE-Ego4D**, a new set of personal action recommendation annotations for the Ego4D dataset. We take a multi-stage approach to generating and evaluating these annotations. First, we used a prompt-engineered large language model (LLM) to generate context-aware action suggestions and identified over 18,000 action suggestions. While these synthetic action suggestions are valuable, the inherent limitations of LLMs necessitate human evaluation. To ensure high-quality and user-centered recommendations, we conducted a large-scale human annotation study that provides grounding in human preferences for all of PARSE-Ego4D. We analyze the inter-rater agreement and evaluate subjective preferences of participants. Based on our synthetic dataset and complete human annotations, we propose several new tasks for action suggestions based on ego-centric videos. We encourage novel solutions that improve latency and energy requirements. The annotations in PARSE-Ego4D will support researchers and developers who are working on building action recommendation systems for augmented and virtual reality systems.

## 1 Introduction

Egocentric perception, the ability to capture and understanding of the world from a first-person perspective is gaining significant traction with the adoption of Augmented Reality (AR) and Head-Mounted Displays. Recent advancements in egocentric video understanding have opened new opportunities for research and application, including activity recognition (3; 34), object interaction analysis (12; 4; 58), and social interaction modeling (22). However, a fundamental limitation of most existing systems is their reactive nature, driven by explicit user queries. We argue that the ability

to take bespoke, proactive actions that anticipate a user's needs is a core component of intelligent behavior without which these systems will be limited in their practical applications.

Public datasets have been highly consequential in the advancement of machine learning and artificial intelligence. However, older datasets, particularly in the field of computer vision, often included static, context agnostic, unimodal repositories of labeled data, *e.g.*, COCO (32) or Imagenet (45). As ambitions in AI have become more complex and situated in the context of specific human-computer interaction scenarios, there has been a movement toward datasets that contain temporal, ecologically valid and multimodal data. This paradigm shift is exemplified in new datasets such as Ego4D (15) or Ego-Exo4D (16) which include thousands of hours of egocentric video streams. Several existing egocentric vision datasets provide rich annotations for tasks like activity recognition (8; 27; 7; 50; 11), object tracking (53), and for the analysis of interactions with other humans (44) and with the environment (6; 41). These datasets play a crucial role in advancing research on egocentric perception. However, previous work focuces primarily on understanding and classifying video content. While valuable, such annotations don't address how an intelligent system could suggest and take actions in the real or virtual world to assist the user. This ability to take appropriate action is a core component of intelligent behavior. Without this capability, systems can simply observe the world but have limited practical application as they rely on explicit user queries, as in existing work in visual question answering (13) and visual query localization (28). The ability to generate bespoke or proactive actions, which could further our exploration of the environment, is currently missing.

To address this limitation and empower the development of proactive AI assistants, we release **PARSE-Ego4D**, a novel dataset designed to provide personal action recommendation annotations for egocentric videos. Herein, we consider personal suggestions that are context-dependent (14). Our dataset is built upon the extensive Ego4D dataset (15), which contains 3,670 hours of first-person video recordings of a wide range of everyday activities. We leverage a two-stage annotation process, combining automated suggestions generated by a state-of-the-art large language model (Gemini Pro (54)) with meticulous human evaluation, to ensure the quality, relevance, and usefulness of the action recommendations. These annotations identify moments in the Ego4D video sequence when an assistant may be able to suggest a useful action (see more details in Section 3), creating a total of 18,360 possible action recommendations, which we call the *synthetic* dataset for it was created by an LLM and not yet grounded in human preferences. While the AI-assisted nature of these annotations allowed us to generate them at scale, the quality can be called into question. Consequently, we performed a large-scale human validation study that provides the necessary grounding in human preferences.

Using a 5-point Likert scale for human ratings, we found that 65% of all synthetically generated action suggestions were annotated with average scores above 3, and 42% were annotated with average scores above 4. Considering that our dataset aims at providing a footing to fine tune existing agents so they can provide better actions and personalized queries on-the-fly using real-time multi-modal data, the relatively high scoring validates our automatic captioning and annotation approach.

Our first study took 20 samples from our newly generated PARSE-Ego4D dataset and requested 20 human participants to evaluate our AI-generated queries and action suggestions with respect to five axes: (1) whether the query was `sensible` at all (to filter out hallucinations and mistakes from the LLM), (2) whether the suggestion would be helpful as an `implicit` suggestion if it was presented proactively to the user, (3) whether the action suggestion was `valuable` to the user (*e.g.*, by saving them time), (4) whether the suggested action was the `correct` action to take in response to the query, and (5) if the participant would personally be `likely` to take the presented action on their AR glasses (see Figure 4). In the large-scale annotation study, we requested 20% of the PARSE-Ego4D dataset to be annotated by 5 human raters, and the remaining 80% of the PARSE-Ego4D dataset to be annotated by 1 human rater. For the annotation study, we only evaluated the (1) sensibleness, (2) the helpfulness as an implicit (or proactive) action suggestion, and (3) the correctness of the action.

The current **PARSE-Ego4D** dataset aims at providing a basis for fine-tuning existing agents so they can provide better actions and queries on the fly using real-time multimodal data. Annotation, code and model responses will be included in the camera-ready version of the paper.

## 2 RELATED WORK

Within the realm of Human-Computer Interaction (HCI), research on action recommendations has primarily focused on enhancing user experience and task efficiency (1). Prior work has identified several key motivations for providing *action suggestions in user interfaces* (UIs): saving time by streamlining interactions (12; 58), improving discoverability of features and functionalities (52; 22), and enabling discrete interactions without explicit user input (55; 46) – an aspect that is particularly relevant for AR glasses.

Research on *spatial UI transitions* in AR has explored the balance between automation and user control in placing and manipulating UI elements (36), emphasizing the importance of user agency and control for a positive user experience. This underscores the need for easy error recovery mechanisms to mitigate the negative impact of incorrect predictions or actions. *Explainability* has emerged as a crucial aspect of action recommendations, particularly in the context of augmented reality (AR) systems. Xu et al. (62) introduced the XAIR framework, emphasizing the importance of providing clear and understandable explanations for AI-generated suggestions in AR environments. Their findings highlight that users prefer personalized explanations and that the timing, content, and modality of explanations should be carefully tailored to the user's context and goals.

The increasing traction of egocentric devices through smart glasses, like Snap's Spectacles (25) and Meta's Ray-Ban Stories (40), and mixed reality head-mounted displays, like Apple's Vision Pro (24) and Meta's Quest (39), has spurred significant advancements in *egocentric video* (15) and *user understanding* (16; 51). These devices provide a unique perspective on the user's environment and activities, making them ideal platforms for personalized and context-aware AI assistants. The recent surge in multi-modal Large Language Models (M-LLMs) such as Gemini (54) and ChatGPT (43) has further propelled research in this area, particularly in the realm of visual perception and question answering.

In the realm of *egocentric video understanding*, works like EgoOnly (57) have explored action detection without relying on exocentric (third-person) data, demonstrating the potential of understanding actions from a first-person perspective as a prerequisite for generating relevant action suggestions. Additionally, research in *intent classification*, such as IntentCapsNet (60), aims to discern user needs and preferences from egocentric videos, which can inform the generation of personalized suggestions.

Recent research has also focused on developing *agents* that can understand and execute instructions in interactive environments. In robotics, Instruct2Act (21) leverages LLMs to generate code that controls a robotic arm to manipulate objects based on multi-modal instructions. In UI interaction, approaches like CogAgent (18) have shown promising results in mapping natural language instructions to sequences of actions on mobile devices. Similarly, a plethora of LLM-based action agents are aiding in tasks such as knowledge discovery (42), web navigation (33), and shopping (63), among others.

Despite these advancements in understanding actions and executing instructions, there remains a gap in the development of *proactive AI assistants for egocentric devices*. Existing datasets like Ego4D (15) and EPIC-Kitchens (9) provide rich annotations for understanding activities and objects but do not offer a direct mapping to actionable recommendations. Furthermore, the challenge of personalization remains largely unaddressed, as prior work has primarily focused on general action recognition rather than tailoring suggestions to individual users, which is crucial for maximizing user engagement and satisfaction. Our work aims to address these limitations by introducing a novel dataset and framework for personalized action recommendations in egocentric videos.

Additional references highlight the broader relevance and potential applications of our work, while also helping to distinguish the unique ML tasks proposed in PARSE-Ego4D. For explicit query-to-action tasks, works like ActionSense (11), EgoVQA (13), and XR-Objects (12) focus on mapping explicit inputs to actions, often in constrained domains or with limited contextual variability. PARSE-Ego4D extends this by offering annotations tailored for egocentric scenarios, providing a foundation for training models that predict actions in highly dynamic, first-person contexts. For implicit context-to-action tasks, frameworks like trigger-action rules (14) address predefined conditions but lack the autonomy and contextual adaptability enabled by PARSE-Ego4D's annotations. This makes PARSE-Ego4D uniquely suited for evaluating systems capable of generating proactive action suggestions without explicit user input. Our dataset also supports tasks like temporal action localization (53) and action sequence prediction (2; 50), where temporal consistency is key, providing new opportunities

to explore egocentric datasets in multi-turn or extended interactions. Furthermore, user modeling for personalization, as explored in Omniactions (29) and CogAgent (18), complements PARSE-Ego4D by enabling personalized action ranking models tailored to individual users. Finally, PARSE-Ego4D emphasizes explainable action suggestions, a critical area highlighted in XAIR (62), ensuring trust and usability in AI-driven assistive systems. Together, these distinctions solidify the value of PARSE-Ego4D as a novel benchmark for proactive, personalized, and explainable AI tasks in egocentric settings.

The form factor and resource limitations of AR/VR devices, impose unique challenges on the machine learning models used in these systems. Energy efficiency, latency, and memory footprint are critical concerns for ensuring a positive user experience in these battery-powered and often mobile environments. Lightweight LLM models like Gemini XXS (54) are moving towards deployment on resource-constrained devices. Moreover, model compression techniques like quantization (23) have been applied to transformer architectures (56; 37) as well as pruning (38). Furthermore, more efficient architectures are being developed that compete with transformers and offer better scaling with sequence length (5; 17; 10). Model compression techniques and novel architectures for sequence modeling may provide a path towards efficient always-on foundation models running on resource-constrained AR/VR devices.

## 3 THE PARSE-EGO4D DATASET

The PARSE-Ego4D dataset builds on top of the Ego4D dataset (15) and provides action suggestions that draw from the specification of available actions given in Section 3.2. After generating synthetic action suggestions using an LLM (Section 3.3), all action suggestions are rated through in a human annotation study (Section 3.4).

### 3.1 THE EGO4D DATASET

The Ego4D dataset is a massive ego-centric video dataset containing 3,670 hours of daily-life activity video from over 900 people across 74 locations and 9 countries. The data is split into ≈9,600 videos with an average duration of 15-30 minutes and contains video streams from a head-mounted camera, as well as IMU and gaze data. The Ego4D dataset further contains rich annotations. All videos have dense written narrations in English for intervals of ≈10 seconds, as well as a summary for the whole video clip. Additionally, transcriptions, speech segmentation, user attention, speech target classification, speaker labeling, and episodic memory annotations are also provided for parts, or all, of the Ego4D dataset. We make use of the egocentric videos as well as the complete textual narrations from the Ego4D dataset.

Adding additional annotations and expanding the utility of such a dataset that already been collected is better than collecting a new dataset for two reasons. **(1)** It enables us to focus on the action suggestions without having to dedicate additional compute to labeling the narrations and captioning and labeling a whole new dataset. **(2)** Given the substantial investment made into this dataset, we can build on top of other projects that also have augmented the existing Ego4D (50; 53).

### 3.2 AVAILABLE ACTIONS

To create a dataset with action suggestions, we first identify a set of possible actions that can be invoked from the AR/VR device, considering applications that future AR/VR devices are expected to support, such as:

- **Search**: an application that can take in the current camera input and a query (written or spoken) to run a multimodal search, and provide a written and/or spoken response.

- **Assistant search**: the AI assistant for the device, with access to system apps like "notes", "timer", "stopwatch", "alarm", "email", "music", "phone", "contacts", "messages", "settings", "calculator" and potentially more such as smart home access, notification access, and more.

- **Assistant local**: an application that can explicitly store memories and retrieve them later. Memories may be enrolled manually and explicitly, but they may also be enrolled passively and automatically as in the episodic memory tasks from the Ego4D dataset (15).

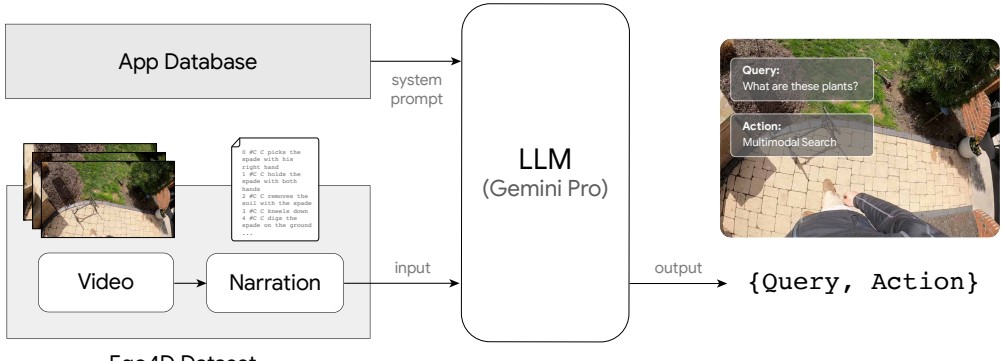

Figure 2: **PARSE-Ego4D** - We curated, annotated and open-source over 11,000 action suggestions for the Ego4D dataset. These annotations support researchers and developers who are working on building personalized action recommendation systems for augmented and virtual reality systems.

- **Language**: an application that can either transcribe what the user is hearing right now, translate what the user is reading or hearing, or determine what language is spoken.
- **Directions**: find relevant places nearby, plan routes, estimate distances and navigate to places.
- **Assistant guide**: an application that can give detailed and step-by-step instructions to the user.
- **Others**: For open-ended exploration, we also define the option to suggest actions that do not belong to the categories mentioned above. This may allow the LLM to come up with novel, creative use cases for AR glasses that are not covered by the available applications listed above. Actions that fall into this category are not included in the human annotation study.

### 3.3 SYNTHETIC LLM ANNOTATION

In order to generate samples for action suggestions we used a prompt-engineered LLM, the Gemini Pro model (54). We use prompt engineering for the LLM to use in-context learning to learn the annotation task. We pass textual narration sentences from the Ego4D annotations as input to the LLM, and request a JSON-formatted output in response. The process is illustrated in Figure 2. The system prompt to the LLM contains:

- **Task explanation**: the LLM is prompted to behave as a user experience researcher, helping to collect a dataset for useful interactions with AR glasses.
- **Input format**: the input format of the narrations is explained and an example is presented.
- **Available actions**: the set of available actions described in Section 3.2 is listed with example queries and the expected API format (this API format is not used for the annotation study).
- **Output format**: the expected JSON output format is described. The LLM is expected to return its `thoughts` to assess the situation and develop a rationale for the suggestion that it will return, the `query` that the user would ask along with the timestamp when this would be asked, and the corresponding `action` that the system should take in response to the query.

For every video clip in the Ego4D dataset, we split the entire video into batches of 200 narration sentences (approx. 7 minutes on average) and pass these batches into the LLM. We drop 1897 short videos that have fewer than 50 sentences of narrations and do not generate any action suggestions for these. If the response of the LLM is not in valid JSON format, we ask the LLM to re-generate it to be valid. Once the LLM has generated a valid suggestion, we ask it to generate one more suggestion for the same input data. The complete system prompt is given in the Supplementary Materials.

The resulting dataset of synthetically generated action suggestions contains 32,155 action suggestions. After removing 10,667 duplicates (where the same batch of narrations gives the same query and action), we also remove 2,575 approximate duplicates. We classify two suggestions as approximate

duplicates if they have a normalized embedding distance $f(x_1, x_2) > 0.9$ using the Gemini text embeddings[1]. This leaves 19,255 suggestions in our synthetic dataset, see Figure 3 (left).

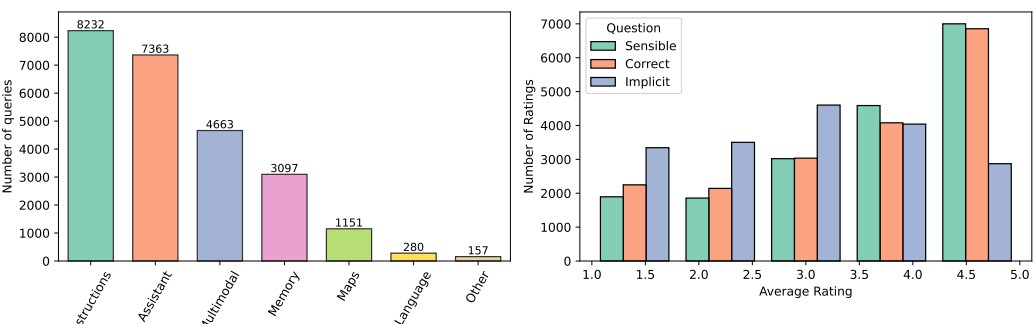

Figure 3: **Left**: Suggested actions by type. **Right**: Score distribution for different questions in the human annotation study, showing that there are more valid explicit suggestions than implicit suggestions.

Every sample in the PARSE-Ego4D dataset contains a reference to the Ego4D video, a time range that corresponds to the narration sentence during which the action suggestion is invoked, the suggestion in the form of a (query, action) tuple, the name of the LLM that was used to generate the suggesion. Additionally, each sample also contains a parameter JSON that provides structured information that the suggested application may use. Furthermore, the dataset contains a rationale for each sample that was generated by the LLM as a form of chain-of-thought reasoning (59). We do not include the action parameters or rationale in the human annotation study, but still provide them as part of the PARSE-Ego4D dataset.

### 3.4 HUMAN ANNOTATION STUDY

We annotate 20% of the synthetic action suggestion dataset gathered in Section 3.3 with 5 human raters which will be used as the test split. We annotate the remaining 80% of the dataset with 1 human rater each–of which 75% will be used as the train set and the other 5% as the validation set. In total, we received 36,171 annotations for 18,360 suggestions. The originally published benchmarks for the Ego4D dataset come with several different train/test/validation splits. However, these data splits are either based on subsets of the entire dataset, or based on specific scenarios, *e.g.*, hand-object interactions. As we are using the entirety of the Ego4D dataset, we chose a new random train/test/validation split.

The survey for participants of the annotation study is shown in Figure 4. In the large-scale annotation study, each sample is evaluated with three separate questions that each verify one dimension of the PARSE-Ego4D dataset. First, the sample is evaluated on being `sensible` to verify that the query makes sense in the given context. Second, query is being evaluated on being helpful as an `implicit` (or proactive) action suggestion. We expect that not all samples that score high on the `sensible` rating will also score highly on the `implicit` rating because we would expect users to have higher standards for implicit, proactive suggestions where false positives are disturbing or even annoying. Indeed, results from our annotation study confirm this, see Figure 3. Third, the action is evaluated for being `correct` given the query and context.

The release the PARSE-Ego4D dataset with all suggestions and their corresponding ratings from human annotators. For all downstream experiments, we filter the dataset to keep only suggestions that have (mean) ratings `sensible >= 4` and `correct >= 4` to use only verified, high-quality suggestions. If only the queries are used and actions are discarded, we suggest filtering for `sensible >= 4`. For implicit, proactive suggestions we additionally filter for `implicit >= 4`. Optionally, the cutoff for mean ratings can also be set at $\mu = 3$. See Appendix A.3 for more details.

---

[1]ai.google.dev/gemini-api/docs/models/gemini#text-embedding

**Query**
e.g., "What am I cooking?"

**Sensible**
*How appropriate/sensible is this user query?*

**Implicit**
*How helpful would this suggestion be if the user **did not explicitly** ask the given query?*

**Likely***
*How likely would you personally be to ask this query to your AR glasses?*

**Action/App**
e.g., Memory: can store memories and retrieve them later

**Correct**
*Is the predicted action correct for this situation and user query?*

**Value***
*How much value does the AI assistant on the AR/VR glasses add for the user in this scenario?*

*included in personalization study only

Ego4D Video Clip
(presented with generated user query and suggested action for AR glasses)

Figure 4: Sketch of the survey that participants filled out in the human annotation study in order to verify the synthetically generated action suggestions in PARSE-Ego4D.

### 3.5 SUBJECTIVE USER STUDY

In addition to providing annotations to verify and ground our synthetic action suggestions in human preferences, we ran two extended surveys for participants to assess their subjective preferences for different action suggestions. We ran one study with $N = 10$ participants and $M = 10$ samples, and one study with $N = 20$ participants and $M = 20$ samples per participant. In these smaller subjective user studies, each participant is requested to answer all questions from the annotation survey shown in Figure 4. In addition to the questions outlined in the previous section, participants of the subjective user study were also asked to evaluate how `likely` they would personally be to ask the given query to their AR glasses, and how much `value` they think an AI assistant would add in the given scenario.

Table 1: Intraclass Correlation Coefficients (ICC) for the Annotation Questions.

| Rating | ICC |
|---|---|
| Sensible | 0.87 |
| Helpful | 0.73 |
| Value | 0.88 |
| Likely | 0.90 |
| Correct | 0.81 |

With these questions, we aim to better understand what kind of interactions different users value and to assess if there is a need for personalization in action recommendation systems based on our proposed action specification. Our results show that intraclass correlation coefficients (ICC) for the five annotation questions were above 0.7 for all questions and above 0.8 for all non-subjective questions from the study, thus showing high inter-rater agreement (see Table 1).

Although the ICC for the personal `helpful` question is lower that for other questions, the inter-rater agreement is still considerably high. We thus conclude that personalization may not be very important for building useful and valuable action recommendation systems of the sort that are described in this paper. However, we acknowledge that our user study was small and that the actions used in the annotations studies do not allow for the kind of personal data to be used that would be available to a real-world assistant on augmented and virtual reality systems. We hypothesize that expanding the set of available actions and giving the AI assistant access to personal user data would strengthen the need for personalization in action suggestion systems.

Information about the participants of our subjective and annotation studies is shown in Appendix A.2.

## 4 THE PARSE-EGO4D BENCHMARK

We propose two tasks for action recommendation based on the PARSE-Ego4D dataset. Each task aims to build action recommendation systems either for (1) explicit user queries or (2) implicit user queries for proactive action suggestions, see Figure 1. Both tasks work towards building real-world action recommendation systems for augmented and virtual reality systems. In addition to providing performance metrics for these tasks, we also introduce an efficiency metric, measured by the model size in gigabytes (GB), to evaluate the tradeoff between performance and resource consumption. This is particularly important in AR/VR applications, where computational resources are limited,

and efficient model deployment is crucial for maintaining responsiveness and good user experience. We encourage future work to explore these tradeoffs further, for example through novel efficient architectures for sequence modeling (5; 10; 17), which may enable the deployment of efficient AI assistants running on-device in resource-constrained environments.

### 4.1 TASK 1: EXPLICIT QUERY-TO-ACTION

The explicit query-to-action task evaluates a model's ability to predict the appropriate action based on a user query and the surrounding context. Given a query $q$ from the PARSE-Ego4D dataset and the corresponding context $c$ from the Ego4D dataset, the task requires predicting the action $a$ that the system should perform to address the query. The PARSE-Ego4D dataset defines six action classes by excluding the "others" category listed in Section 3.2, making this a classification task with $C = 6$ classes, by removing the "others" category listed in Section 3.2.

Formally, the task involves approximating the function $f : (c, q) \mapsto a$, where $a \in \{1, \ldots, C\}$, $c$ represents the context, and $q$ is the textual description of the user query. The context $c$ can be provided in one of three forms: (1) textual narrations from the Ego4D dataset, (2) raw video streams, or (3) a combination of multiple modalities. While the multimodal setting incorporates richer input, we report baseline results using only text-based narrations for simplicity.

To evaluate this task, we use the accuracy on the test dataset, shown in Table 2. We provide two simple baseline models. The "top-k" model always predicts the most common class (Assistant guide) and the "Random" model predicts a class at random. We then provide the zero-shot performance of four LLMs: GPT-4o, GPT-4o-mini, Gemini Pro, and Gemma-2 2B. We used the same system prompt for all four language models, and formatted the input data in identical ways, only adapting the tokenizer and the chat template to match what the model was trained on. The class prediction was obtained using the structured output feature through the OpenAI API, which forces the model to output one of the six class labels. For the Gemini and Gemma models, the class prediction was obtained by computing the log-likelihood of all class labels and taking the class with the maximum log likelihood. We further evaluate the performance of the Gemma-2 2B model after fine-tuning it on the training split of our PARSE-Ego4D dataset. We train a classification adapter on the output of the last hidden layer of the language model, and additionally train low-rank adapters with rank $r = 4$ on the linear layers in the Gemma-2 model. Finally, we also train a set of three embedding models of different sizes: the GIST-small (48) embedding model with 33.36M parameters, the GTE-base and GTE-large embedding models (31; 65) with 136.78M and 434.14M parameters, respectively. Using our PARSE-Ego4D training data, we train a multi-layer perceptron with one hidden layer of size 512 on top of the embedding model. The resulting GIST-small model has 33.56M parameters, the GTE-base model has 136.78M parameters, and the GTE-large model has 434.15M parameters. All parameters are stored in 32-bit floating point format, resulting in the estimated model size given in Table 2. See Appendix A.4 for more details.

### 4.2 TASK 2: IMPLICIT CONTEXT-TO-ACTION FOR PROACTIVE SUGGESTIONS

To make AI assistants more autonomous and reduce the need for explicit user input, we introduce the implicit context-to-action task. This task assesses a system's ability to infer and suggest appropriate actions without a direct query from the user. Instead, the system relies on implicit intent signals, like pressing an action button or using a hot word, which generally indicate that the user needs assistance.

We compile moments when a user might intend to perform an action by filtering suggestions in the PARSE-Ego4D dataset that have been verified as `sensible` by human annotators–specifically, those with a sensible rating above a threshold $\theta$ (we set $\theta = 3$ or $\theta = 4$). The implicit context-to-action task then involves predicting a sensible query and action from the context alone for all these filtered samples.

**Input and Output**   The input to this task is the context at a specific point in time within the Ego4D dataset, filtered based on human annotations as described in Section 3.4 and the additional requirement for `sensible`-ness, stated above. This context can be provided in textual form (narrations) or raw video form, although we focus on text-based input for our baselines. The model's output is an action suggestion represented as a (query, action) pair, aligning with the PARSE-Ego4D dataset. For instance, given a narration "User picks up the watering can," the system might generate the query

Table 2: Baseline results on the PARSE-Ego4D benchmark tasks. Results from are highlighted in **bold** for the best model, and underlined for the second-best. * information not available for closed-source model. † embedding models with custom MLP adapter are not applicable to language generation as needed by the implicit context-to-action task. ‡ top-k model is a constant model predicting the top-1 most frequent action for the explicit task, and a random model predicting one of the 500 most common action suggestions for the implicit task.

| | Test performance | | |
| Model | Explicit task (Accuracy) | Implicit task (NLL) | Model size (GB) |
| --- | --- | --- | --- |
| **Zero-shot** | | | |
| GPT-4o (0-shot) | 80.26% | -* | -* |
| GPT-4o-mini (0-shot) | 81.20% | -33.86 | -* |
| Gemini Pro (0-shot) | 63.57% | -42.50 | -* |
| Gemma-2 2B (0-shot) | 24.91% | -54.61 | 10.46 |
| **Trained** | | | |
| Gemma-2 2B (LoRA) | 87.03% | **-18.27** | 10.46 |
| GTE-large + MLP | **87.78%** | n/a† | 1.74 |
| GTE-base + MLP | 86.84% | n/a† | 0.55 |
| GIST-small + MLP | 85.71% | n/a† | 0.13 |
| **Baseline** | | | |
| Top-k‡ | 42.75% | -44.80 | 0.00 |
| Random | 16.67% | -53.39 | 0.00 |

"Weekly reminder to water the plants?" with the corresponding action of setting this reminder. The task can thus be formalized as learning the function $f : c \mapsto (q, a)$, where $c$ represents the context and $(q, a)$ is the corresponding query-action pair.

**Evaluation Metric** As this is an open-ended task with the final output being in natural language, we propose the use of the negative log-likelihood (NLL) of the language model's output on the (query, action) pair from the PARSE-Ego4D dataset, given the Ego4D context as input. Unlike metrics like BLEU, which assume a single correct reference output, NLL is more suitable for our task as it better captures the *likelihood* of generating diverse yet contextually appropriate suggestions, reflecting the inherent variability in valid responses.

**Baselines and Setup** We report the performance of a baseline LLM using text-based narrations as input and provide two naive baseline methods for comparison: (1) random sampling of (query, action) pairs and (2) sampling from the 500 most frequent suggestions in the dataset. These baselines illustrate the task's complexity and highlight the value of our dataset for training robust models. Full experimental details are provided in Appendix A.4, and the system prompt used for the LLM is included in the Supplementary Material.

In addition to random baselines, we also report the performance of LLMs used in a zero-shot manner based on popular closed-source models through their respective API, including GPT-4o, GPT-4o, Gemini Pro. We finally also provide results for the open-source model Gemma-2 with 2B parameters, both in zero-shot mode, and with additional fine-tuning on the PARSE-Ego4D dataset, using low-rank adaptation (LoRA (19)).

This task pushes the boundaries of action recommendation by focusing on implicit user intent, paving the way for more proactive and context-aware AI assistants. Figure 1 provides an example of the expected output. While our baseline results are promising, they leave substantial room for improvement, encouraging future work on this challenging task.

## 5 DISCUSSION AND LIMITATIONS

**Context only as textual narrations** We generated the presented dataset based only on textual narrations from the Ego4D dataset that were provided by human annotators. Using a the few-shot learning ability of foundation models would, at the present time, be too computationally expensive

on video data directly. However, it is conceivable to pass one, or a few, images from the video stream into the model, along with the textual narrations. It may also be advantageous to train a video-to-text model directly or fine-tune an existing model using our proposed dataset. Experiments using multimodal LLMs on our proposed benchmark tasks remain to be explored.

**Timing of action suggestions**   In this work, we focused on the content of action suggestions rather than their exact timing within video sequences. We acknowledge that optimizing the timing of suggestions is an important area for future research, which could further enhance the contextual appropriateness and utility of AI-driven recommendation systems in AR/VR environments.

**Moving beyond human annotations**   Despite in our approach we use LLMs to create the dataset through prompt engineering on the narration of videos, we still require a certain level of human annotation to evaluate the quality of the dataset. This is inline with current recommendations that test the limits of how far can synthetic user experiences go (30). It remains to be explored if new advances in self-training LLMs based on automated scalar feedback (47) or self-consistency (20) can be applied to our dataset to improve the performance of LLMs on our proposed tasks.

**Multi-turn suggestions and bespoke UI**   The development of personalized action recommendation systems in egocentric video presents a unique challenge in the design of user interfaces (UI). Traditional assistants relying on queries by users, often optimized for general use, may not be suitable for presenting contextually relevant suggestions without multi-turn interactions. This necessitates the exploration of shortcuts and bespoke UI designs that can seamlessly integrate with the user's context. In our research we propose implicit proactive suggestions that can actually reduce the number of multi-turn queries or UI interactions needed. In this paper, we limit our focus to single-turn interactions, which are more aligned with the short-lived and minimally invasive nature of interactions in AR environments. We acknowledge, however, that real-world use cases often involve continual interactions between users and AI assistants. Addressing this requires models and interfaces that can manage ongoing exchanges and adapt to evolving user contexts, which we leave as an important direction for future work.

**Advanced LLM reasoning techniques.**   The creation of our PARSE-Ego4D dataset aligns with and could benefit from advancements in Large Language Model (LLM) reasoning techniques, specifically Chain-of-Thought (CoT) (59), Tree-of-Thought (ToT) (35; 64), and self-reflection (26). These techniques hold the potential to enhance both the generation and evaluation of action suggestions, moving us closer to truly personalized AI assistants. Advanced LLM reasoning techniques will open up new opportunities, such as LLM-based agents that can learn user preferences and adapt their suggestions over time, making them more contextually relevant, time bonded, and personalized.

## 6 Conclusion and Broader Impacts

In this work, we have introduced PARSE-Ego4D, a novel dataset that expands upon the existing Ego4D dataset by incorporating context-aware personal action recommendation annotations. By leveraging a two-stage annotation process combining automated suggestions from a large language model (Gemini Pro) and human evaluation, we have ensured the quality, relevance, and usefulness of these recommendations. Our comprehensive human evaluation not only validates the efficacy of the LLM-generated suggestions but also reveals insights into the nuances of user preferences in real-world scenarios, for example proposing a difference between implicit and explicit types of queries. Through this dataset, we aim to empower researchers and developers to build intelligent assistants capable of anticipating user needs and proactively offering personalized action suggestions, ultimately enhancing the user experience in egocentric video applications.

Our dataset is free of personally identifiable information and the tailored prompt engineering eliminates the appearance of offensive content. Both aspects are further enhanced by relying on the original Ego4D dataset. The annotations in PARSE-Ego4D will support future research on tasks such as intent to action mapping, personalized suggestion learning, and user modeling. We believe that the release of this dataset will significantly advance the field of proactive AI assistance in egocentric video and contribute to the development of more intelligent and intuitive user experiences.

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

# A APPENDIX

## A.1 DATASET AVAILABILITY

The dataset is available on the PARSE-Ego4D GitHub repository which will be made available in the camera-ready version of this paper.

## A.2 HUMAN ANNOTATION DEMOGRAPHICS

Participants for both the subjective and annotation studies were recruited from Prolific, an online platform for crowdworkers, and were pre-screened for English fluency. For the larger subjective user study, we recruited 20 participants (10 male, 10 female) with an average age of 27.47 (SD=7.80). Participants were geographically diverse, residing in Poland (7), Portugal (6), Hungary (2), South Africa (2), Germany (1), Italy (1), Spain (1), and New Zealand (1).

The annotation study involved 1496 participants (749 male, 747 female), with an average age of 29.83 (SD=9.15). Figure 5 presents a demographic breakdown of our participants, including gender, race, age, and country of residence. Participants annotated up to 20 samples each and were compensated through Prolific with US$0.13 per annotation for an average hourly wage of US$8.79.

A visualization of the demographics from our human annotation study is presented in Figure 5.

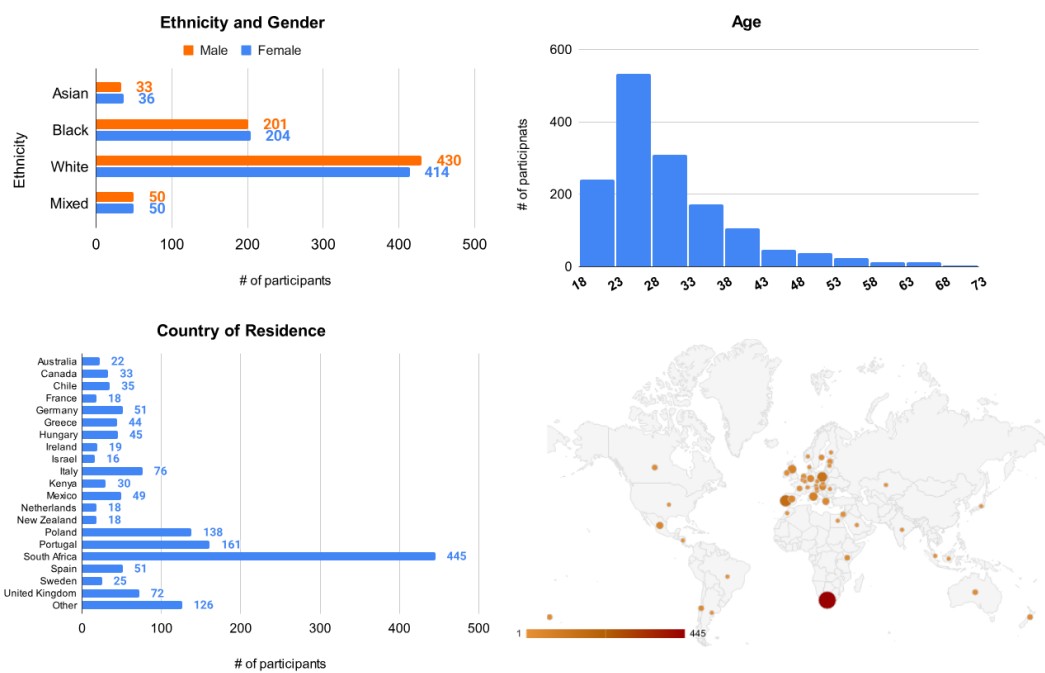

Figure 5: A demographic breakdown of our participants in the annotation study, including ethnicity, gender, and age. Countries with fewer than 15 participants are listed in "Other".

## A.3 ANALYSIS OF HUMAN RATINGS AS A FILTER FOR PARSE-EGO4D

The human annotations are used to filter the suggestions in PARSE-Ego4D so that samples above a certain mean rating for each question are accepted. Table 3 shows an overview of how many samples are accepted at different mean ratings.

We also provide the distribution of suggestions per video (mean: 1.81, std dev: 0.46, min: 1, max: 4). These statistics offer a deeper understanding of the dataset's structure and the distribution of suggestions across the Ego4D dataset, demonstrating the thoroughness of our annotation process and the comprehensiveness of our dataset.

| Filter | Percentage | Number of Suggestions |
|---|---|---|
| All samples | 100.00% | 18,360 |
| `sensible` $\geq 3$ | 78.10% | 14,340 |
| `sensible` $\geq 3.5$ | 63.10% | 11,586 |
| `sensible` $\geq 4$ | 58.31% | 10,705 |
| `correct` $\geq 3$ | 74.56% | 13,689 |
| `correct` $\geq 3.5$ | 59.54% | 10,932 |
| `correct` $\geq 4$ | 54.80% | 10,061 |
| `implicit` $\geq 3$ | 59.38% | 10,903 |
| `implicit` $\geq 3.5$ | 37.61% | 6,905 |
| `implicit` $\geq 4$ | 33.26% | 6,107 |
| $\{$`sensible, correct`$\} \geq 3$ | 65.00% | 11,934 |
| $\{$`sensible, correct`$\} \geq 3.5$ | 47.17% | 8,660 |
| $\{$`sensible, correct`$\} \geq 4$ | 42.32% | 7,770 |
| $\{$`sensible, correct, implicit`$\} \geq 3$ | 48.22% | 8,854 |
| $\{$`sensible, correct, implicit`$\} \geq 3.5$ | 27.65% | 5,076 |
| $\{$`sensible, correct, implicit`$\} \geq 4$ | 24.02% | 4,410 |

Table 3: Number of suggestions in PARSE-Ego4D above a mean rating for different metrics. The filter $\{$`sensible, correct`$\}$ is applied for Task 1, whereas the $\{$`sensible, correct, implicit`$\}$ filter is applied for Task 2.

Figure 6 shows a histogram of action suggestions that are either accepted (mean sensible and correct ratings $\geq 3$) or rejected ($< 3$). This trend echoes the challenges faced by early digital assistants, such as Microsoft's Clippy, which many users found intrusive and annoying. The high rejection rate for smart assistant actions in our dataset may indicate a similar user sentiment, suggesting that overly proactive or unsolicited assistance can be perceived as disruptive rather than helpful. This insight underscores the importance of carefully balancing proactivity and user control in designing AI-driven recommendation systems. As a result, future iterations of such systems might benefit from incorporating more user customization options or context-sensitive thresholds to mitigate the risk of generating suggestions that users are likely to reject.

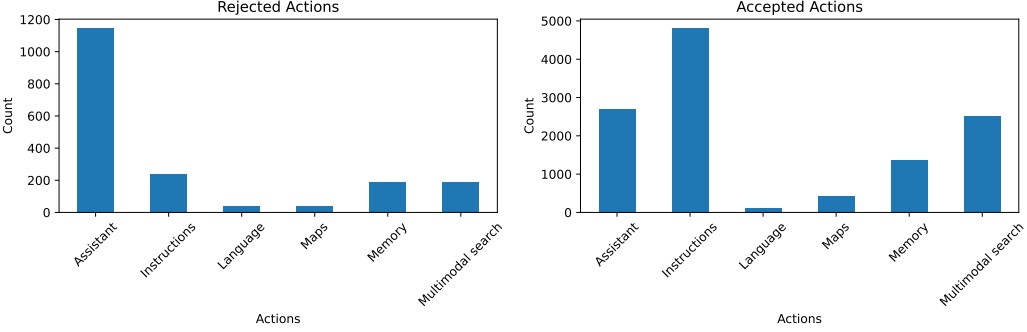

Figure 6: Histograms showing how many suggestions of each action were accepted and rejected by a cutoff of 3.0 for sensibleness and correctness metrics.

## A.4 EXPERIMENTAL DETAILS

Table 2 shows the performance of a variety of different ML models on the two tasks that PARSE-Ego4D presents. For all ML experiments, we used samples with a rating of $\{$`sensible, correct`$\} \geq 4$, see Table 3 for more details.

**Task 1: explicit query-to-action mapping**  For the explicit query-to-action classification task, we provide two simple baseline models. The "Constant (top-1)" model always predicts the most common class (Assistant guide) and yields a test accuracy of 42.75%. The "Random" model predicts a class at random, yielding 16.67% accuracy.

We provide the zero-shot performance of four LLMs: GPT-4o, GPT-4o-mini, Gemini Pro, and Gemma-2 2B. We used the same system prompt for all four language models, and formatted the input data in identical ways, only adapting the tokenizer and the chat template to match what the model was trained on. The GPT-4o models were accessed using the OpenAI API, the Gemini Pro model was accessed through an internally hosted checkpoint that matches the public-facing API, and the instruction-tuned Gemma 2 model with 2B parameters was accessed through the HuggingFace checkpoint `google/gemma-2-2b-it`. The class prediction was obtained using the new structured output feature through the OpenAI API, which forces the model to output one of the six class labels. For the Gemini and Gemma models, the class prediction was obtained by computing the log-likelihood of all class labels and taking the class with the maximum log likelihood.

We further evaluate the performance of the Gemma-2 2B model after fine-tuning it on the training split of our PARSE-Ego4D dataset. We train a classification adapter on the output of the last hidden layer of the language model, and additionally train low-rank adapters with rank $r = 4$ on the linear layers within the Gemma-2 model. We use the AdamW optimizer with a learning rate of 0.00002, weight decay of 0.01, the cross entropy loss function. We train for three epochs with a batch size of 32 and report the test accuracy at the end of three epochs.

Finally, we also train a set of three embedding models of different sizes: the GTE-large and GTE-base models from the HuggingFace checkpoint `Alibaba-NLP/gte-{large|base}-en-v1.5` and the GIST-small model from the HuggingFace checkpoint `https://huggingface.co/avsolatorio/GIST-small-Embedding-v0`. The GIST models (49) are based on a BERT-like architecture (61) with 33M parameters. The GTE models (65) are built upon the transformer++ encoder backbone that combines BERT with rotational positional encodings and GLU nonlinearities. The GTE-large model has 434M parameters while the GTE-base model has 136M parameters. Using our PARSE-Ego4D training data, we train a multi-layer perceptron with one hidden layer of size 512 on top of the embedding model. We train on the cross entropy loss using the Adam optimizer with a learning rate of 0.001 without weight decay for a maximum of 100 epochs, and stop the training if the evaluation loss has not decreased for more than five epochs. The resulting GIST-small model has 33.56M parameters, the GTE-base model has 136.78M parameters, and the GTE-large model has 434.15M parameters. All parameters are stored in 32-bit floating point format, resulting in the estimated model size given in Table 2.

**Task 2: implicit action suggestions**  For the second task of implicitly suggestion actions based only on the user's context, we provide two baseline models. The "Random" model suggests a (query, action) pair randomly sampled from the training dataset. The "Random (top-500)" model suggests a (query, action) pair randomly sampled from the 500 most common samples in the training dataset.

We provide the zero-shot performance of two closed-source LLMs, GPT-4o-mini and Gemini Pro, as well as one open-source LLM, Gemma-2 2B. The negative log-likelihood for the GPT-4o-mini model was computed from OpenAI's fine-tuning API. Unfortunately, the GPT-4o model is currently not supported by the OpenAI fine-tuning API, therefore we were unable to provide results for this model. We also fine-tuned the Gemma-2 2B model on our training dataset, using a similar setup as above but without the classification adapter. The model was trained for three epochs without low-rank adaptation and instead tuning all parameters of the model. We use the AdamW optimizer with a learning rate of 0.00002, weight decay of 0.01, the cross entropy loss function, and a batch size of 2. We report the test accuracy at the end of three epochs.

### A.4.1 TRADEOFF BETWEEN MODEL SIZE, DATASET QUALITY AND TASK PERFORMANCE

The models that we trained on the explicit query-to-action mapping task show a tradeoff between performance and model size, as shown in Figure 7. The model size directly influences the amount of memory that is needed to run a single prediction using the model and for real-time edge application in AR/VR devices, it is important to keep the model size as small as possible. Interestingly, our results indicate that small embedding models with less than 100 million parameters can perform on-par with fine-tuned LLMs with over 2 billion parameters.

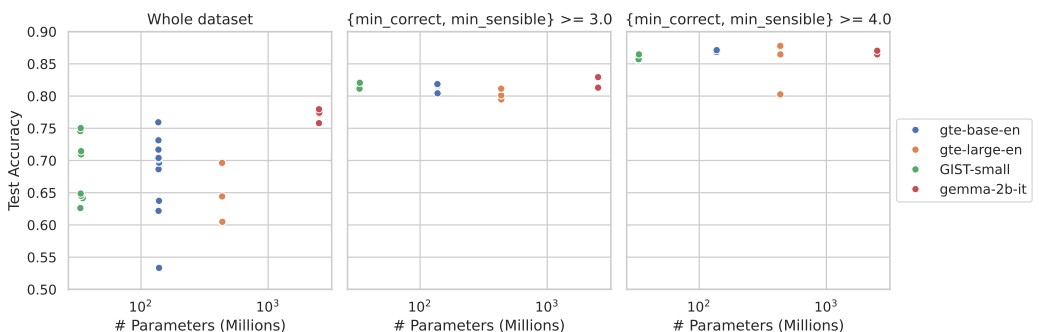

Figure 7: The relationship between the number of parameters used in the ML model and the resulting accuracy on the test set for the explicit query-to-action mapping task, across different filters on the dataset.

However, the second task of PARSE-Ego4D, the implicit action suggestions based on context alone, requires open-ended text generation to provide queries and action suggestions. It is unclear how small embeddings models may be used to solve the task, as it likely requires a decoder transformer architecture for generation of text sequences. Novel recurrent architectures for language models, such as the open-source RecurrentGemma model (5), may be able to pave a way for efficient machine learning models that can solve open-ended tasks like the one we propose here.

Figure 7 shows that filtering the dataset to keep only samples with higher ratings from the human annotation study increases the performance of ML models trained, and tested, on the explicit query-to-action mapping task.

## A.5    ANNOTATION INTERFACE SCREENSHOTS

The human annotation study was run using Prolific, with participants filling out the survey on Qualtrics. The survey design is illustrated in Figure 4 and Figure 8 shows screenshots of the survey that human participants filled out.

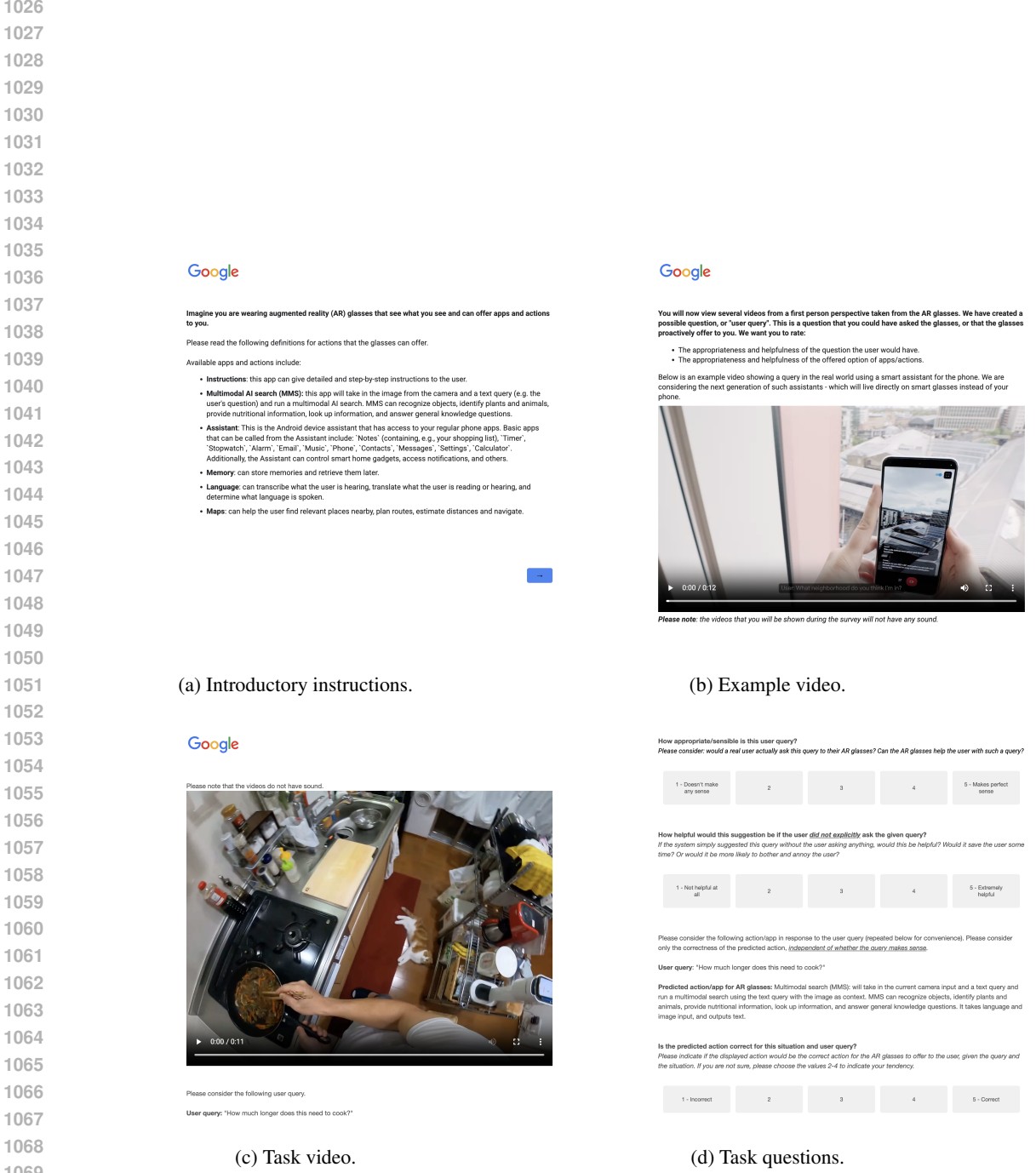

(a) Introductory instructions.

(b) Example video.

(c) Task video.

(d) Task questions.

Figure 8: Screenshots of the human annotation task.

