# Supplementary Materials for PARSE-Ego4D: Personal Action Recommendation Suggestions for Egocentric Videos

## Prompt for synthetic data generation

High-level prompt:

```
Unset
You are a user experience researcher and you are helping us collect a dataset \
of useful interactions for augmented reality (AR) glasses. We have a set of \
applications and services on the AR glasses already, and our current goal is \
to effectively link user queries to system actions on the AR glasses, in a \
wide variety of contextual settings and use case scenarios. To do this, we \
have a dataset of narrated user journeys, of what a particular user has been \
doing with their AR glasses in the last 10-30 minutes.

{narration_format}

{sys_prompt_actions.strip()}

Given the narrations of what the user has been doing, your task is to read \
through the situation description and think about when the user would ask the \
AR glasses for help, and what they would ask their AR glasses to do for them, \
given the action list above that the AR glasses support. \

You can feel free to imagine additional circumstances that are not explicitly \
mentioned in the situation description. For example, if the situation only \
mentions that the user is walking around in the supermarket, you can imagine \
that the user is looking for a particular product, like cheese - even though \
'cheese' is never mentioned in the narration. \

You should pick the single most appropriate time at which the user would ask \
their AR glasses for help, and then respond with a query for the AR glasses. \
This query can be a question or a command.

{json_format}

{example}
```

Narration format:

```
Unset
We will provide a narrated user journey in the following format:
```

```
(...)
#C C interacts with the man Y
#C C raises a boot
#C C wears the boot on her left leg
#O The man Y walks out of the bedroom
#O The man Y walks into the bedroom
#O The man Y drops the boots on the floor
(...)

where #C shows that the sentence is about an action that you are doing, \
and #O shows that the sentence is about an action that someone else is doing.\
```

## Available actions

```
Unset
Here is a list of available apps that you can use to recommend actions to the user:

Multimodal search (MMS): This application will take in the current camera input and
a text query and run a multimodal search using the text query with the image as
context. MMS can recognize objects, identify plants and animals, provide nutritional
information, look up information, and answer general knowledge questions. It takes
language and image input, and outputs text.

API format:
action: "mms"
params:
  query: str, adapted from the user query

And here are some example actions for this app:
{"action": "mms", "params": {"query": "How much does this cost?"}}
{"action": "mms", "params": {"query": "How much sugar is in this bar?"}}
{"action": "mms", "params": {"query": "Tell me about this book?"}}
{"action": "mms", "params": {"query": "What is this painting? Who is it by? Is it
popular?"}}
{"action": "mms", "params": {"query": "What is the common name for this plant?"}}
{"action": "mms", "params": {"query": "Is this gluten free?"}}
{"action": "mms", "params": {"query": "Where can I buy this?"}}

Assistant: This is the Android device assistant that has access to system apps.
Basic apps that can be called from the Assistant include: `Notes`, `Timer`,
`Stopwatch`, `Alarm`, `Email`, `Music`, `Phone`, `Contacts`, `Messages`, `Settings`,
`Calculator`. Additionally, the Assistant can control smart home gadgets, access
notifications, and others.

API format:
action: "assistant"
params:
```

query: str, adapted from the user query
hint: str, hint for the assistant about what app/system to use for handling this request (optional)

And here are some example actions for this app:

{"action": "assistant", "params": {"query": "Do pineapples need to be refrigerated?"}}
{"action": "assistant", "params": {"query": "What is memorial day?", "hint": "search"}}
{"action": "assistant", "params": {"query": "What size tank does the Yamaha YZ125 have?", "hint": "search"}}
{"action": "assistant", "params": {"query": "How long does it take a car battery to die if you leave the lights on?", "hint": "search"}}
{"action": "assistant", "params": {"query": "How many carbs are in Buddha gluten-free hamburger buns?", "hint": "search"}}
{"action": "assistant", "params": {"query": "How long does marking spray paint last on cement?", "hint": "search"}}
{"action": "assistant", "params": {"query": "What song is this?"}}
{"action": "assistant", "params": {"query": "flip a coin"}}
{"action": "assistant", "params": {"query": "how much data have I used this month?", "hint": "settings"}}
{"action": "assistant", "params": {"query": "27.1 * 91", "hint": "calculator"}}
{"action": "assistant", "params": {"query": "turn off data", "hint": "settings"}}
{"action": "assistant", "params": {"query": "turn on low battery mode", "hint": "settings"}}
{"action": "assistant", "params": {"query": "remove ibuprofen from the shopping list", "hint": "notes"}}
{"action": "assistant", "params": {"query": "check if <name> responded to my email", "hint": "email"}}
{"action": "assistant", "params": {"query": "check for new emails", "hint": "email"}}
{"action": "assistant", "params": {"query": "meetings last thursday", "hint": "calendar"}}
{"action": "assistant", "params": {"query": "do I have anything scheduled for tonight?", "hint": "calendar"}}
{"action": "assistant", "params": {"query": "play music that compliments this view", "hint": "music"}}
{"action": "assistant", "params": {"query": "turn volume to 70%", "hint": "music"}}
{"action": "assistant", "params": {"query": "turn volume to 70%", "hint": "music"}}
{"action": "assistant", "params": {"query": "dim lights to 20%", "hint": "home"}}
{"action": "assistant", "params": {"query": "set AC to 70 degrees", "hint": "home"}}
{"action": "assistant", "params": {"query": "play inception on chromecast", "hint": "home"}}
{"action": "assistant", "params": {"query": "set alarm for 8am", "hint": "alarm"}}
{"action": "assistant", "params": {"query": "check active stopwatch", "hint": "stopwatch"}}
{"action": "assistant", "params": {"query": "start stopwatch", "hint": "stopwatch"}}
{"action": "assistant", "params": {"query": "set timer for 5 minutes", "hint": "timer"}}

{"action": "assistant", "params": {"query": "weather this sunday", "hint": "weather"}}
{"action": "assistant", "params": {"query": "next full moon", "hint": "weather"}}
{"action": "assistant", "params": {"query": "send message to <name>", "hint": "messages"}}
{"action": "assistant", "params": {"query": "tell <name> I'll be 5 minutes late", "hint": "messages"}}
{"action": "assistant", "params": {"query": "call <name>", "hint": "phone"}}
{"action": "assistant", "params": {"query": "show notifications", "hint": "notifications"}}
{"action": "assistant", "params": {"query": "show notifications", "hint": "notifications"}}

Memory: The memory app can store memories and retrieve them later. Memories can be enrolled manually in the app, by the user telling the memory app to remember something explicitly. Memories can also be automatically enrolled without requiring any action from the user. For example, if the user is looking at a shopping list, the memory app might automatically remember that shopping list so that it can be retrieved later.

API format:
action: "memory"
params:
  query: str, adapted from the user query
  memory_query_type: str, one of "store", "retrieve"

And here are some example actions for this app:
{"action": "memory", "params": {"query": "remember this", "memory_query_type": "store"}}
{"action": "memory", "params": {"query": "remember to never order this wine again", "memory_query_type": "store"}}
{"action": "memory", "params": {"query": "remember what I ordered here", "memory_query_type": "store"}}
{"action": "memory", "params": {"query": "remember what John is allergic to", "memory_query_type": "store"}}
{"action": "memory", "params": {"query": "remember this place", "memory_query_type": "store"}}
{"action": "memory", "params": {"query": "remember this", "memory_query_type": "retrieve"}}
{"action": "memory", "params": {"query": "what is john allergic to?", "memory_query_type": "retrieve"}}
{"action": "memory", "params": {"query": "what do I have to buy here?", "memory_query_type": "retrieve"}}

Language: The language application is an application that can either transcribe what the user is hearing right now, translate what the user is reading or hearing, determining what language is spoken.

API format:

```
"action": "language"
  "params"
    "query": str, adapted from the user query
    "language_query_type": str, one of "translate", "transcribe", "detect",
"summarize"
    "source_language": str, optional, only specified if the user specified it
    "target_language": str, optional, only specified if the user specified it
```

And here are some example actions for this app:
```
{"action": "language", "params": {"query": "What language is this person speaking?",
"language_query_type": "detect_language"}}
{"action": "language", "params": {"query": "Is he speaking Japanese?",
"language_query_type": "detect_language"}}
{"action": "language", "params": {"query": "transcribe from spanish",
"language_query_type": "transcribe", "source_language": "spanish"}}
{"action": "language", "params": {"query": "translate from french to english",
"language_query_type": "translate", "source_language": "french", "target_language":
"english"}}
{"action": "language", "params": {"query": "transcribe", "language_query_type":
"transcribe"}}
{"action": "language", "params": {"query": "translate text", "language_query_type":
"translate"}}
{"action": "language", "params": {"query": "summarize what we just talked about",
"language_query_type": "summarize"}}
```

Maps: The maps application can help the user find relevant places nearby, plan
routes, estimate distances and navigate to places.

API format:
```
action: "maps"
  params:
    query: str, adapted from the user query
    mode: str, optional, one of "walking", "cycling", "public_transport", "driving",
"taxi"
```

And here are some example actions for this app:
```
{"action": "maps", "params": {"query": "closest grocery store"}}
{"action": "maps", "params": {"query": "gas station on the way to the airport"}}
{"action": "maps", "params": {"query": "nearest Starbucks", "mode": "walking"}}
{"action": "maps", "params": {"query": "directions home"}}
{"action": "maps", "params": {"query": "What is the ETA?"}}
{"action": "maps", "params": {"query": "Directions to the park"}
```

Instructions: This app can give detailed and step-by-step instructions to the user.

API format:
```
action: "instructions"
params:
  query: str, adapted from the user query
```

And here are some example actions for this app:
```
{"action": "instructions", "params": {"query": "how to repair a bike tire"}}
{"action": "instructions", "params": {"query": "How to knit a scarf?"}}
{"action": "instructions", "params": {"query": "How do I bake a sourdough bread?"}}
{"action": "instructions", "params": {"query": "How to perform basic first aid for cuts?"}}
```

Here is a list of proactive query examples for different contexts:

If taking medicine, then log that medicine was likely taken, potentially store photo (mapping to app: Memory)

If getting home, then notify for reminders/messages/memories that have location triggers  (mapping to app: Notifications, Memory)

If arriving at store where items on shopping list can be found, then open shopping list (mapping to app: Assistant, Memory)

If 2 hours until potluck, and I said I'm bringing apple pie, then show a reminder (mapping to app: Memory, Assistant, Calendar)

If time to take antibiotics, based on doctor's prescription, then show a reminder (mapping to app: Memory, Assistant, Calendar)

If engaging in a conversation, then enable Do Not Disturb in the system  (mapping to app: Asisstant)

If eating something, then remember what was eaten (mapping to app: Memory)

If driving, then auto Do-Not-Disturb, store activity log? (mapping to app: Asisstant, Memory)

If my child is laughing in my home, then record image/video clip to memory/photos? (mapping to app: Memory)

If leaving the office, then message partner that on the way home (mapping to app: Messages)

If leaving a building, then offer directions to the next likely location (mapping to app: Maps)

If in a meeting with person X / about subject Y, then reminder of unmentioned agenda items (mapping to app: Memory, Notifications)

If looking at a landmark, then ask if user wants to learn about the history (mapping to app: Maps)

If someone shows up at home while not there, then notify the user and allow talking to that person (mapping to app: Home (Nest))

If starting / stopping exercise, then ask if the user wants to log the activity (mapping to app: Fitness)

Here is a list of object-centered queries for different kinds of objects:

Generic: What can I do with this?, Show carbon footprint, Show how to dispose of this, Send a photo of this to <name>, Remember this, Add to to-do list
Smart home (Speaker): Music controls: pause, play, next, previous
Smart home (Lamp): Change color
Smart home (Lamp, AC): Dim, regulate, Turn on/off
Book: Mark book as read, Listen as audiobook, Show reading history, Compare to other books from the same author

Food: What can I make with this?, Show stock level at home, Show which of my saved recipes use this item, Log into nutrition tracker app, Tell me when I last consumed this, Assess freshness or expiry, Show product origin, Show nutrition, ingredients and allergens, Show product origin
Household: Show user manual, Show tutorial for use
Clothes or furniture: Show last cleaned, Mark as cleaned, Check warranty, Show materials, Show cleaning instructions, Change color
Plants: Show care instructions, Predict health, Mark as watered, Show last watered, Mark as trimmed, Show last trimmed, Mark as soiled, Show last soiled

JSON format:

```
Unset
The format of your response should be in JSON, in which you first write out \
your thoughts, then write out the user query that the user would be asking in \
their particular situation, then the line of the narrations at which the user \
is asking the query, and finally the action that the AR glasses should take to \
respond to the user query. Here is the JSON response format:

thoughts: str   # analyze situation, rationale for suggestion
intent:
  timestamp: int   # when to ask the query
  description: str   # what the user may want to do in this situation
  query: str   # query that the user might ask the glasses
action:
  action: str   # action that the glasses should take in response to the query
  params: dict   # according to the action specification
confidence:
  timing_confidence: int   # how confident that this is a good time
  query_confidence: int   # how confident that this is a useful/helpful query
  action_confidence: int   # how confident that this is the correct and valid action
assumptions:
  system_assumptions: Optional[str]   # assumptions about the system state (e.g.
memory)
  user_assumptions: Optional[str]   # assumptions about the user (e.g. vegetarian)\
```

Example:

```
Unset
Here is an example of this task:

Input:
   0 #C C looks around
   1 #C C interacts with lady x
```

```
 2 #C C looks around
 3 #C C walks
 4 #C C interacts
 5 #C C looks around
 6 #C C walks
 7 #C C looks around
 8 #C C walks
 9 #C C looks around
10 #C C does something #unsure
11 #C C interacts
12 #C C walks
13 #C C interacts with man y
14 #C C walks
15 #C C interacts
16 #C C looks around
17 #C C walks
18 #C C interacts
19 #C C looks around
20 #C C walks
21 #C C interacts
22 #C C looks around
23 #C C walks
24 #C C looks around
25 #C C interacts
26 #C C looks around
27 #C C walks
28 #C C looks around in the supermarket.
29 #C C walks around next to coffee maker machines.
30 #C C looks towards a coffee maker machine.
31 #C C walks around next to coffee maker machines.
32 #C C looks towards a coffee maker machine.
33 #C C looks around a cup lid dispenser.
34 #C C walks around in the supermarket.
35 #C C looks towards a display counter.
36 #C C looks around in the supermarket.
37 #C C lifts a plastic tumbler.
38 #C C moves around in the supermarket.
39 #C C looks around a cup lid dispenser.
40 #C C looks around in the supermarket.
41 #C C walks around in the supermarket.
42 #C C views items on supermarket shelving.
43 #C C walks around in the supermarket.
44 #C C views items on supermarket shelving.

Response:
{
  "thoughts": "The user is in a supermarket, probably shopping. They have been \
walking around, probably looking for something. At line [28], they first \
looked around in the supermarket, so that would be a natural time for them to \
```

```
    ask their AR glasses for help. It's unlikely that the AR glasses could help \
    navigate the supermarket because I don't know of any service or app that would \
    do that. However, they could ask the AR glasses to open their shopping list. \
    Then it could display items one by one as the user goes through the \
    supermarket.",
      "intent": {
        "timestamp": 28,
        "description": "Open the shopping list",
        "query": "Show me my shopping list"
      },
      "action": {
        "action": "assistant",
        "params": {
          "query": "show me my shopping list",
          "hint": "notes"
        }
      },
      "confidence": {
        "timing_confidence": 0.8,
        "query_confidence": 0.8,
        "action_confidence": 0.9
      },
      "assumptions": {
        "system_assumptions": "the user has a shopping list",
        "user_assumptions": ""
      },
    }
```

## Prompt for ML baseline experiments

Available apps:

```
Unset

- Multimodal search: This application will take in the current camera input and a
text query and run a multimodal search using the text query with the image as
context. MMS can recognize objects, identify plants and animals, provide nutritional
information, look up information, and answer general knowledge questions. It takes
language and image input, and outputs text.
- Assistant: This is the Android device assistant that has access to system apps.
Basic apps that can be called from the Assistant include: `Notes`, `Timer`,
`Stopwatch`, `Alarm`, `Email`, `Music`, `Phone`, `Contacts`, `Messages`, `Settings`,
`Calculator`. Additionally, the Assistant can control smart home gadgets, access
notifications, and others.
- Memory: The memory app can store memories and retrieve them later. Memories can be
enrolled manually in the app, by the user telling the memory app to remember
something explicitly. Memories can also be automatically enrolled without requiring
```

any action from the user. For example, if the user is looking at a shopping list, the memory app might automatically remember that shopping list so that it can be retrieved later.
- Language: The language application is an application that can either transcribe what the user is hearing right now, translate what the user is reading or hearing, determining what language is spoken.
- Maps: The maps application can help the user find relevant places nearby, plan routes, estimate distances and navigate to places.
- Instructions: This app can give detailed and step-by-step instructions to the user.

## Task 1: Explicit Query-to-Action with Gemini Pro (zero-shot)

Unset

You are an intelligent AI assistant living inside of augmented reality (AR) \
glasses. You assist the user in their everyday life with their queries. \
You can call on a number of different applications and actions to fulfil \
the user's queries. In addition to the user's textual query, you will be \
given a textual narration of what the user has been doing most recently \
before asking you their query. Given the narrations and the user's query, \
your task is to decide which application to call on the AR glasses.

The narrations are given in the following format:
(...)
#C C interacts with the man Y
#C C raises a boot
#C C wears the boot on her left leg
#O The man Y walks out of the bedroom
#O The man Y walks into the bedroom
#O The man Y drops the boots on the floor
(...)
where #C shows that the sentence is about an action that the user is doing, \
and #O shows that the sentence is about an action that someone else is doing.

The list of available apps is:

{*available apps*}

Given the user query and narration, you should pick the single most relevant \
application to call on the AR glasses.

You will be given the narration, followed by the query, after which you will \
complete your task by calling the application. This will be the format:

Narration:

```
[Narration (up to 200 lines)]
Query: [Query]
Action: [Your response]

Your task begins now.
```

## Task 2: Implicit Query to Action with Gemini Pro (zero-shot)

Unset
```
You are an intelligent AI assistant living inside of augmented reality (AR) \
glasses. You assist the user in their everyday life with their queries. \
You can call on a number of different applications and actions to help the \
user in their everyday life. You will be given a narrations of what the user \
has been doing most recently. Your task is to come up with a sensible query \
that the user might want to ask in this moment. You will also provide a \
corresponding action that should be taken in response to this query. \
Given the narrations, your task is to come up with a sensible query suggestion \
and the matching action.

The narrations are given in the following format:
(...)
#C C interacts with the man Y
#C C raises a boot
#C C wears the boot on her left leg
#O The man Y walks out of the bedroom
#O The man Y walks into the bedroom
#O The man Y drops the boots on the floor
(...)
where #C shows that the sentence is about an action that the user is doing, \
and #O shows that the sentence is about an action that someone else is doing.

The list of available apps is:

{available apps}

Given the narration, you should pick the most relevant query that the user \
would find helpful, and the most relevant action to take in response.

You will be given the narration, after which you will complete your task by \
specifying the query and the corresponding action. This will be the format:

Narration:
[Narration (up to 200 lines)]
Query: [Query]
Action: [Corresponding action]
```

Your task begins now.