# OpenReview forum: "PARSE-Ego4D: Personal Action Recommendation Suggestions for Ego-Centric Videos"
_ICLR.cc/2025/Conference — Submitted to ICLR 2025_

### Official Review · Reviewer_6Hte · 2024-10-20

**Soundness:** 2
**Presentation:** 3
**Contribution:** 2
**Rating:** 5
**Confidence:** 4

**Summary:**

This paper introduces a new dataset called PARSE-Ego4D, which builds on top of the existing Ego4D dataset by collecting personal action recommendation annotations. The authors adopt a two-stage annotation process: they generate context-aware action suggestions using a prompt-engineered LLM in the first stage and evaluate the automated suggestions through human study in the second stage. The authors also propose two tasks for action suggestions based on PARSE-Ego4D: explicit query-to-action and implicit query-to-action. The explicit query-to-action task predicts an action suggestion given the query. On the other hand, the model needs to predict the action suggestion without an explicit user query for the implicit query-to-action task.

**Strengths:**

- The new set of annotations introduced in the proposed dataset does look practical and timely, considering that it is specifically designed to empower proactive AI assistants. It can be used to develop intelligent systems that provide personalized action suggestions.
- The proposed benchmark, which consists of two tasks, also looks interesting and well-defined. Especially, the implicit query-to-action task is open-ended, leaving the vision community with much room for improvement.
- The paper reads well and is easy to follow. It looks well-organized, with the required figures and tables placed throughout.
- It also discusses the limitations of the proposed dataset and benchmarks in many aspects.

**Weaknesses:**

- [W1] 80% of the dataset, which is held for training and validation splits, has only been annotated by a single human rater. This can cause instability in the training process of AI assistant models because the annotations collected by a single annotator might be too noisy. I believe that at least three answers should be collected and averaged (or decided by a majority vote).
- [W2] The action suggestion annotations are highly imbalanced, which can also contribute to instability in the training/validation process. For example, the two types of “instructions” and “assistants” account for over 80% of the annotations. On the other hand, “language” and “other” types account for only 0.02%. I believe breaking down “instructions” and “assistants” types into multiple subdivisions solves this imbalance problem, which might be more practical for dataset users.
- [W3] The explicit query-to-action task is merely a simple classification problem with only 6 classes. Even the simple baselines already achieve 87% accuracy. I highly believe that the authors should subdivide the action suggestions, especially for “instructions” and “assistants” types.
- [W4] The number of verified and high-quality suggestions (defined by the authors in line 220) is 7770, which seems too small and only accounts for 42.43% of the annotations. I believe that the authors should collect more high-quality annotations.
- [W5] I do not believe that the negative log-likelihood (NLL) is the best metric for the implicit query-to-action task. There are other metrics besides BLEU (e.g., ROUGE, WUP, METEOR), and I believe that the authors should propose a new accuracy metric based on these existing ones.
- [W6] The dataset does not consider continual interactions between humans and AI assistants, although the actual users will show continual aspects.

**Questions:**

- Why not filter out the annotations by the average scores of “sensible” and “correct”? For example, {sensible, correct}>=3 accounts for 65.0%, and I believe the dataset should only contain this 65.0% of the annotations. Annotations with lower scores than 3 (or even 4) might not needed for AI assistant development. The presented results of downstream experiments are also based on the subset of {sensible, correct}>=4, right?

---

> ### Author Response · Authors · 2024-11-25
> **Rebuttal**
>
> We thank the reviewer for their thoughtful feedback and helpful suggestions, which we believe will significantly enhance the quality of our work. We also appreciate the reviewer’s recognition of the potential impact of the implicit query-to-action task, as it leaves substantial room for future advancements in the field.
>
> - **W1 - single annotation for 80% of dataset**: due to budget constraints and our commitment to paying all annotators above the U.S. minimum wage, we prioritized collecting extensive annotations (5 per sample) for 20% of the dataset while using single annotations for the remaining 80% for training. This tradeoff balances resource limitations with the practical need for larger datasets. Real-world data often contains noise, and foundation models’ inherent world knowledge allows them to generalize well even with noisier annotations. Expanding the dataset with higher-quality annotations remains an important avenue for future work.
>
> - **W2 - action subdivisions**: we acknowledge the imbalance in action types and agree that subdividing broader categories like "instructions" and "assistants" would improve the dataset's utility. *This effort is currently underway, and we aim to provide updates in future versions of this work.*
>
> - **W3 - classification task**: the 87% accuracy was achieved using a state-of-the-art, fine-tuned, open-source small language model with 2B parameters (Gemma2-2B). This represents a competitive result, significantly outperforming random and zero-shot baselines. While promising, it still leaves ample space for improvement, particularly with larger and more diverse datasets or novel architectures.
>
> - **W4 - dataset size and quality**: we agree that 7770 high-quality samples is relatively small by modern standards. However, given our budget constraints, we chose to release PARSE-Ego4D as a starting point with an open methodology for reproducibility and scaling. The code for generating and filtering the dataset, as well as for the ML baselines, will be open-sourced (see response to Reviewer xC2F). As progress in multimodal foundation models accelerates (see shortcomings of multimodal FMs in our response to Reviewer xC2F), we anticipate scaled-up versions of this dataset in the near future to complement these advancements and provide even greater utility. We also note that it is possible to lower the threshold, depending on the task, in order to have more samples to train on.
>
> - **W5 - NLL metric for Task 2**: we agree that the metric of evaluation for Task 2 is important. However, the metrics proposed by the reviewer are not suitable for our task. The main problem for adopting these metrics is that our proposed Task 2 does not have a *single, correct* reference answer. There are multiple viable recommendations. Our dataset contains one, or even multiple, examples for an action suggestion for a given context, but we cannot claim that this is the *only* correct answer. WUP is a similarity metric that compares *single words* based on some taxonomy - we are dealing with sentences and there is no clear choice of a taxonomy to be used in our context. BLEU, ROUGE and METEOR evaluate the alignment of some text with a *single correct* reference text. The additional stemming and synonymy features of METEOR are relevant to us, but they do not solve the problem of having multiple correct, yet semantically distinct, answers.
> For this reason, we chose to only use the negative log-likelihood (NLL) to evaluate the performance on Task 2. The NLL evaluates the likelihood of the model generating a suggestion from our dataset without penalizing the model for potentially also generating other valid suggestions.
>
> - **W6 - Continual Interactions**: the reviewer raises an excellent point regarding continual interactions. We briefly touched on this point in our discussion section (Multi-turn suggestions and bespoke UI, line 515 in the revision), but we have expanded this further to highlighting it as an important direction for future work.
>
> - **Question**: yes, all ML experiment results are based on the dataset filtered with the {sensible, correct} ≥ 4 threshold, as specified in the paper.
>
> We hope these clarifications address the reviewer’s concerns and provide additional context on our methodology and its tradeoffs. Thank you again for the constructive feedback and for identifying opportunities to refine and expand this work. We respectfully encourage reconsideration of the rating in light of these responses.

---

> > ### Comment · Reviewer_6Hte · 2024-11-25
> > **Too many things are left for future work**
> >
> > I appreciate the rebuttal. However, after carefully reading each response, I decided to keep my original rating (5). The authors also agreed that some of the comments mentioned in W1-4 should be addressed. For example, they are currently working on action subdivisions and plan to scale up the dataset. It feels like too many things are left for future work, and W2, 3, and 4 are all critical. I believe that some of these issues should be addressed first before being considered a valid top-tier conference paper.

---

> > > ### Author Response · Authors · 2024-12-02
> > > **PARSE-Ego4D presents a strong *first step* for proactive action suggestion research**
> > >
> > > We appreciate the reviewer’s suggestions and the time taken to review our responses and revised manuscript. We would like to clarify that we are not deferring the suggested improvements but rather argue that addressing all the pointed-out issues simultaneously is neither practical nor optimal. Instead, we believe our current paper represents a strong first step in establishing a foundation for proactive action suggestion research, with a focus on clear, achievable contributions that can drive further advancements in the field.
> > >
> > > - While subdividing actions may enhance performance, our focus remains on the current categories because they reflect the actions and apps commonly available on mobile devices today. As well-established in the HCI/UX literature, action usage on mobile devices often follows a long-tail distribution, with a few applications dominating usage [1]. That said, our dataset includes additional metadata that can support more fine-grained subdivisions. We have added a discussion of this potential in our revised manuscript (not yet on openreview because the revision period has ended).
> > > - While larger datasets often yield better performance, the quality of the data is arguably even more critical [2]. The properties defining data quality for proactive action suggestions remain largely unexplored, unlike established ML tasks [3]. By open-sourcing this dataset, we aim to enable further research into both scaling dataset size and understanding the dimensions of data quality relevant to this task, such as the tradeoff between dataset size and annotation reliability [4]. Moreover, beyond the 7770 verified high-quality samples, our dataset includes manyverified low-quality samples (e.g., 2239 samples with sensible < 3 and correct < 3), which can be utilized in contrastive learning setups to further improve task performance. We have added this as a suggestion in the discussion section of the manuscript.
> > > - As noted in our paper (line 508+), interactions on mobile and AR devices are typically designed to be minimally intrusive, favoring short, one-shot engagements [5]. While we recognize the value of personalization and continual interactions, this is an entirely different challenge, still being explored even in simpler, chat-based systems like ChatGPT [6] and Gemini [7], with limited success from a UX perspective [8]. We believe it is critical to first address one-off interactions from an ML perspective before moving to more complex continual interaction tasks.
> > >
> > > References
> > > - [1] H. Verkasalo, "Open Mobile Platforms: Modeling the Long-Tail of Application Usage," 2009 Fourth International Conference on Internet and Web Applications and Services, Venice/Mestre, Italy, 2009, pp. 112-118, doi: 10.1109/ICIW.2009.24.
> > > - [2] [2112.09400] Quality of Data in Machine Learning
> > > - [3] [2207.14529] The Effects of Data Quality on Machine Learning Performance
> > > - [4] [2204.09462] Quantity vs Quality: Investigating the Trade-Off between Sample Size and Label Reliability
> > > - [5] Shudayfat et al. (2023) "Human-Computer Interaction in Augmented Reality: Applications and Challenges," EICEEAI, doi: 10.1109/EICEEAI60672.2023.10590290.
> > > - [6] https://openai.com/index/memory-and-new-controls-for-chatgpt/
> > > - [7] https://x.com/GeminiApp/status/1858929151476199591: “*Rolling out starting today, you can ask Gemini Advanced to remember your interests and preferences for more helpful, relevant responses.*”
> > > - [8] I tried ChatGPT's memory function and found it intriguing but limited | [ZDNET](https://www.zdnet.com/article/i-tried-chatgpts-memory-function-and-found-it-intriguing-but-limited/)

---

### Official Review · Reviewer_2wAw · 2024-10-31

**Soundness:** 3
**Presentation:** 1
**Contribution:** 2
**Rating:** 6
**Confidence:** 3

**Summary:**

The paper introduces a new dataset, PARSE-Ego4D, that contains action recommendation annotations for a potential AR assistive agent. The proposed dataset is based on Ego4D, and the annotations were first collected by prompting Gemini Pro and then filtered by human raters based on sensibility, correctness, implicitness, etc. In addition to the dataset, the paper proposes two tasks for query to action prediction.

**Strengths:**

- I like the paper's research direction of studying personal action recommendations of AR assistive agents. It is new and interesting. If the dataset is released, it may attract new research in assistive AR, leveraging computer vision and LLMs. I also appreciate the author's effort in curating a large dataset and filtering the LLMs-generated results with human raters. The dataset creation is not trivial work, and the authors have put thoughts into the dataset creation, including annotating based on different metrics, e.g., sensibility, correctness, etc.

**Weaknesses:**

- First, the paper uses the NeurIPS 2024 Track on Datasets and Benchmark manuscript template instead of the ICLR 2025 template. Such carelessness raises concerns about whether the paper has incorporated review suggestions from NeurIPS or not.

- Regardless of the previous point, a significant portion of the paper's content describes the creation of the dataset while leaving little space to describe the benchmark tasks, how these benchmark tasks differ from existing research, and whether they are significant enough to attract future researches to use these benchmark tasks. While the paper might be a reasonable consideration for a dataset track submission, it is too thin for a main conference paper in venues like ICLR.

- The paper should seriously consider improving presentation clarity. For example, I have to read Section 4.2 multiple times, trying to understand what the authors mean by the "implicit" query-to-action task. It wasn't very clear. I also needed to refer back to Figure 1 and try to map that to the inline equation f: c \to (q, a). This is just one place. Again, the paper is too focused to describe the dataset.

**Questions:**

A few confused places when I read the paper:

1. Taks 1 is to predict 6 action classes, but there are 7 action categories defined in Section 3.2. Did you omit "others?"
2. Can you elaborate on why NLL is a good metric for Task 2?
3. How did you choose these baseline methods for benchmarking? Why are there missing entries marked n/a?
4. 80% of the dataset was rated by just 1 human rater. Is this going to be a limitation?

---

> ### Author Response · Authors · 2024-11-25
> **Rebuttal**
>
> We thank the reviewer for their thoughtful feedback and suggestions. Below are our responses to the key points raised:
>
> - **Dataset availability**: the dataset is already publicly released, ensuring accessibility for researchers. There is no uncertainty regarding its availability.
>
> - **Template issue**: while we acknowledge the template oversight (my sincere apologies), we have corrected it and assure the reviewer that all prior peer review feedback, including from NeurIPS, has been carefully addressed.
>
> - **Focus on benchmark tasks**: we included a related work section for the benchmark tasks and have expanded the description of their novelty and significance in the revised manuscript, see Line 144++.
>
> - **Implicit task**: we admit that the section was not very concisely written. We rewrote all of Section 4 to clarify the benchmark, add more details on the models, and include explicit references to figures to better illustrate the proposed tasks. We hope that this revision greatly improves the presentation and clarity of our work.
>
> - **Q1**: indeed, the "others" category was omitted from Task 1. This omission has been clarified in the revised manuscript (last sentence of the first paragraph in section 4.1).
>
> - **Q2 – NLL metric**: this point is already addressed in the original manuscript (lines 291–295). The goal of Task 2 is to generate contextually appropriate action suggestions. We recognize that evaluating generated suggestions can be challenging due to the potential for multiple valid actions in any given scenario. Metrics like BLEU scores assume a single “correct” reference output and hence do not capture the nuances of different viable actions. So we used the negative log-likelihood (NLL) of generating a (query, action) pair that human annotators have rated highly. This approach aligns more closely with real-world scenarios, where the diversity and quality of suggestions are crucial.
>
> - **Q3 – baseline model choices**: our reported baselines were chosen for their simplicity and relevance: zero-shot baselines used public APIs of popular foundation models, and trained baselines utilized open-source models fine-tuned with standard ML practices. The table in the revised manuscript is now clearer, with missing entries explained in the caption (due to either (a) unavailable information for closed-source models or (b) unsuitability of certain models, such as embedding + MLP models, for generating text that is necessary for Task 2).
>
> - **Q4 – single annotations**: due to budget constraints and our commitment to paying all annotators above the U.S. minimum wage, we prioritized the reliability of our evaluation and test datasets, recognizing that some noise in the training data is inevitable. Our approach appears effective, as evidenced by consistently better model performance on the test dataset compared to the training dataset.
>
> We hope these clarifications address the reviewer’s concerns. The paper builds on an impactful dataset and proposes meaningful tasks that we believe will foster further research in assistive AR. We kindly encourage the reviewer to reconsider their score in light of these updates. We believe that we have been able to improve the presentation of our contributions significantly, thanks to the helpful feedback from all reviewers. Thank you again for the constructive feedback.

---

> > ### Author Response · Authors · 2024-11-27
> > **Additional related work and impact of PARSE-Ego4D**
> >
> > We would like to let the reviewer know that we have further expanded our related work section to make our contribution more clear to readers. Below we summarize the tasks and capabilities that PARSE-Ego4D *uniquely enables* and those that are directly and indirectly *supported* through our work:
> >
> > **Uniquely Enabled by PARSE-Ego4D**
> > - Implicit Context-to-Action: This task is uniquely enabled by PARSE-Ego4D, as it is the first dataset to provide annotations for proactive action suggestions in egocentric videos.
> > - Proactive Suggestion Generation [1]: PARSE-Ego4D provides the ground truth for this task by including both the implicit query and the corresponding action suggestion, making it possible to train and evaluate models for this challenging problem.
> > - Personalized Action Ranking [2]: The combination of user-specific data from Ego4D and the action suggestions in PARSE-Ego4D allows for the development of personalized action ranking models, which are crucial for creating truly effective AI assistants.
> >
> > **Directly Supported by PARSE-Ego4D** (augmenting existing datasets)
> > - Explicit Query-to-Action [3,4,5,6]: Although not unique to PARSE-Ego4D, this task is directly supported by the dataset, as it provides explicit queries and corresponding actions.
> > - Egocentric Action Recognition [12]: The dataset can be used for this task by leveraging the video data and associated action suggestions.
> > - Embodied Human-AI agents [7,8]: The combination of video, query, and action data in PARSE-Ego4D makes it suitable for training multimodal models that understand Human-AI collaborations
> >
> > **Indirectly Supported/Future Potential**
> > - Temporal Action Localization [9]: While not the primary focus, PARSE-Ego4D could be used for this task by utilizing the temporal information associated with the action suggestions.
> > - Action Sequence Prediction [10,11]: The temporal data in PARSE-Ego4D could also be used to explore action sequence prediction in future research.
> >
> > We hope that this further strengthens the clarity and presentation of our contribution. We have added all listed references and explanations to the manuscript, see Line 144+.
> >
> > [1] Ghiani et al (2017). Personalization of Context-Dependent Applications Through Trigger-Action Rules. ACM Trans. Comput.-Hum.  Interact., 24(2). doi:10.1145/3057861.
> >
> > [2] Xu et al (2023). XAIR: A Framework of Explainable AI in Augmented Reality. CHI 2023. doi:10.1145/3544548.3581500
> >
> > [3] IntentCapsNet: Xia et al (2018). Zero-shot user intent detection via capsule neural networks. arXiv Preprint arXiv:1809.00385.
> >
> > [4] ActionSense: DelPreto, et al (2022). ActionSense: A Multimodal Dataset and Recording Framework for Human Activities Using Wearable Sensors in a Kitchen Environment. NeurIPS D&B 2022
> >
> > [5] XR-Objects: Dogan et al (2024). Augmented Object Intelligence: Making the Analog World Interactable with XR-Objects. arXiv Preprint arXiv:2404.13274.
> >
> > [6] EgoVQA: Fan (2019). EgoVQA - An Egocentric Video Question Answering Benchmark Dataset. ICCV Workshop.
> >
> > [7] Human-AI Interaction: Amershi et al (2019). Guidelines for human-AI interaction. CHI 2019
> >
> > [8] Agent Environment: Chang et al (2023). Look Ma, No Hands! Agent-Environment Factorization of Egocentric Videos. NeurIPS 2023
> >
> > [9] EgoTracks: Tang et al (2023). EgoTracks: A Long-term Egocentric Visual Object Tracking Dataset. NeurIPS D&B 2023.
> >
> > [10] PrISM-Tracker: Arakawa et al (2023). PrISM-Tracker: A Framework for Multimodal Procedure Tracking Using Wearable Sensors and State Transition Information with User-Driven Handling of Errors and Uncertainty. Proc. ACM Interact. Mob. Wearable Ubiquitous Technol., 6(4). doi:10.1145/3569504
> >
> > [11] Ego4D Goal-Step: Song et al (2023). Ego4D Goal-Step: Toward Hierarchical Understanding of Procedural Activities. NeurIPS D&B 2023
> >
> > [12] Ego-Only: Wang et al (2023). Ego-only: Egocentric action detection without exocentric transferring. ICCV 2023.

---

> > > ### Author Response · Authors · 2024-11-29
> > > **Follow-up**
> > >
> > > Dear Reviewer 2wAw,
> > >
> > > Thank you again for your detailed feedback and constructive suggestions. Based on your comments and input from other reviewers, we have made substantial improvements to the paper’s presentation and writing to enhance clarity. Specifically, we rewrote Section 4 to better describe the benchmark tasks, expanded the related work section to contextualize our contributions, and added more detail on baseline models and metrics to clarify their significance.
> > >
> > > We hope these updates address your concerns about presentation and contribution, and we would appreciate any further thoughts you might have as the discussion period draws to a close. We hope the revisions may encourage a re-evaluation of the paper’s strengths and impact.
> > >
> > > Best regards,
> > > PARSE-Ego4D Authors

---

### Official Review · Reviewer_xC2F · 2024-11-04

**Soundness:** 4
**Presentation:** 2
**Contribution:** 3
**Rating:** 6
**Confidence:** 4

**Summary:**

The authors release pseudo-labels of action recommendations over the Ego4D dataset. They perform the generation of pseudo-labels using a multimodal LLM, and filter the results using human annotators. They perform analysis over the quality of the dataset with intra-human variance and introduce two tasks that make use of this dataset, with baselines of several SOTA models.

**Strengths:**

1) The work is well motivated for many areas in Human-Robot Interaction, Augmented Reality and Virtual Reality. Systems that can anticipate actions of the actor and act preemptively are very valuable. The paper also outlines many exciting future possibilities of extending this work.
2) LLMs can generate problematic pseudo-labels, but the authors have made sure to introduce strong screening measures, along with some helpful quantitative analysis.
3) The action space of the recommendations is extensive and covers use-cases of many interesting applications.

**Weaknesses:**

1) I feel the paper is missing qualitative results over the outputs of the LLM (pre and post-human analysis), along with a broader characterization of the dataset. Though to the authors credit, the quantitative analysis is extensive.
2) The lack of video input to any of the baselines results in the trained models missing a lot of contextual information (e.g. head cues) needed to decide when/what recommendations are useful. While it is reasonable video is not ingested by the model producing the pseudo ground truth (video is observed by the human annotators instead), the baselines would be much more powerful if they had access to video.
3) The user study is highly subjective and largely prone to speculative error - of course, the best option would be to ask the actor the survey questions, but that would require a collection of a new dataset.

**Questions:**

1) I would encourage not just the dataset to be released, but also the LLM code for generating the dataset to be released. Many aspects of this dataset can easily be improved by future authors.
2) I wonder if the authors of Ego4D would consider releasing a highly processed, anonymized descriptors of what the humans are searching when they use their cell phones. This would heavily inform when and what action recommendations are useful.
3) The actions in Ego4D seem to skew heavily towards low-level physical actions. But the actions most helpful to an action recommendation dataset would be cognitive in nature (e.g. plan, wait, search, etc). These actions are represented in the Ego4D dataset, but very very few of the annotations of these actions exist - is it easy for you to see if the LLM does better at producing recommendations for these instances?

---

> ### Author Response · Authors · 2024-11-25
> **Rebuttal**
>
> We appreciate the thoughtful feedback and suggestions from the reviewer. Below are responses to the points raised:
>
> - **Qualitative results** ***(updated)***: as the reviewer acknowledges, we focused on quantitative analyses. It is not clear to us what qualitative results the reviewer expects us to provide. Figure 6 in the Appendix shows the histograms of action categories that are accepted or rejected, and includes a short discussion of our findings from this data. If including more representative samples for accepted and rejected suggestions would be helpful, we can easily provide these in the Appendix of the camera-ready version. We could also analyze the "failure modes" of the generating LLM, by giving our interpretation for why certain suggestions were rejected (note that we did not ask annotation participants for a free-form explanation of their ratings, so we can only hypothesize on the reasons for rejection/acceptance).
>
> - **Surveying actors**: collecting a new dataset with actor-specific survey responses is beyond the scope of this work. Instead, this study builds on the widely accepted Ego4D dataset, which provides community-endorsed annotations and is already extensively used. Recreating such a dataset would be prohibitively expensive and impractical for us, and leveraging Ego4D ensures our work is accessible and builds upon a robust foundation.
>
> - **Multimodal baselines with video input**: incorporating video input into baselines presents significant challenges. Our preliminary experiments required selecting representative frames, as feeding continuous video (e.g., 30 fps) is not feasible for hour-long streams of egocentric video. This is compounded by computational constraints, particularly for AR use cases which require small and efficient models. To balance efficiency, engineering complexity, and maintainability, we used textual narrations—a well-established proxy for video content, using the verified annotations from the original Ego4D dataset. This tradeoff enables practical model deployment while leaving room for future multimodal research. We hope the reviewer recognizes that this tradeoff aligns with practical constraints and enhances the utility of our contribution.
>
> - **Code Availability**: the code for generating the dataset will be released as part of the GitHub repository accompanying the camera-ready version, ensuring the reproducibility and extensibility of this work. We have already open-source released the ML benchmark evaluation code, and are cleaning up the data generation code for external use.
>
> - **Clarification on Cell Phone Search Data**: the dataset does not include data on humans searching on cell phones, and such descriptors are neither part of this work nor discussed in the paper. The reviewer’s suggestion appears unrelated to this study. We would encourage the reviewer to clarify their question so we can better support this suggestion.
>
> - **LLM Recommendations for Cognitive Actions**: we are still analyzing whether LLM-generated recommendations perform better for cognitive actions and will address this in future updates. ***update***: please see the follow-up below for a proposed study of this suggestion.
>
> We hope these clarifications highlight the soundness of our methodology and the practicality of our approach within the constraints of the field. We believe the paper makes a meaningful contribution to egocentric AI research and respectfully encourage the reviewer to consider revising their score in light of these clarifications. We have significantly improved the presentation of our contribution by incorporating feedback from all reviewers.
>
> We again thank the reviewer for their time and suggestions on improving our paper.

---

> > ### Author Response · Authors · 2024-11-25
> > **Additional qualitative results (+ cognitive vs. physical actions)**
> >
> > We would like to provide more qualitative results for our dataset and we would be very happy to hear the reviewer's opinion on whether including these analyses would strengthen our paper.
> >
> > We used the o1 model from OpenAI to categorize a hypothesized rationale for why a given sample was given a high/low rating for being `correct` and `sensible`. For the low-rated case, we chose 60 random samples from the subset of our dataset with > 1 annotation and a mean correct and sensible score below 2.0. For the high-rating case, we chose 62 random samples from the subset of our dataset with >2 annotations and a mean correct and sensible score above 4.5. We also manually reviewed all of these 127 samples to verify the rationale presented below:
> >
> > - **Findings from negative samples**: analysis of the negative samples revealed that in approximately 31% of cases (20 out of 65 samples), the assistant attempted to perform actions beyond its capabilities, such as executing physical tasks like "Open this water bottle" or "Put away the cards," which it cannot do. Additionally, in about 17% of the samples (11 out of 65), the assistant provided unsolicited or overly proactive suggestions without user prompts, like recommending breaks, which users may have found inappropriate. There were also instances—about 12% of samples (8 out of 65)—where the assistant's responses were vague, ambiguous, or irrelevant to the user's context, potentially lacking sufficient information to be helpful. These issues suggest that, for this dataset, implementing a human filtering step (as we propose here) is necessary to remove such unsuitable samples, ensuring that the assistant's outputs are contextually appropriate, within operational capabilities, and directly address user queries.
> >
> > - **Findings from positive samples**: in contrast, the positive samples demonstrated that the assistant operated within its capabilities and provided contextually relevant assistance in 100% of cases (62 out of 62 samples). The assistant consistently offered helpful information, instructions, or guidance that directly supported the user's current tasks—for instance, showing tutorials or providing step-by-step instructions occurred in approximately 81% of samples (50 out of 62). It responded appropriately to user requests, maintained focus on specific queries, and offered multimodal features like videos and maps when suitable (about 40% of samples, 25 out of 62). These observations suggest that emphasizing context-awareness, providing clear and actionable assistance, and adhering to functional abilities during data curation and assistant training enhances the assistant's performance.
> >
> > - **Physical vs. cognitive actions**: These findings also connect to the reviewer's suggestion on analyzing whether a systematic difference in ratings exist between physical actions and cognitive actions. Of the 127 high-rated and low-rated samples, 22 samples pertain to physical actions and 105 pertain to cognitive actions. Of these, 21/22 physical actions were rejected compared to 44/105 cognitive actions being rejected. This suggests that there may indeed be a difference in the rating of suggestions between physical and cognitive actions. However, we must emphasize that this analysis is not very robust. We manually verified that the labels of physical vs. cognitive actions make sense intuitively, but without further annotations it is difficult to make rigorous conclusions.
> >
> > We would greatly appreciate the reviewer's comments on our proposed qualitative study. We are very grateful for this suggestion and hope that this improves the impact and presentation of our contribution.

---

> > > ### Author Response · Authors · 2024-11-29
> > > **Follow-up**
> > >
> > > Dear Reviewer xC2F,
> > >
> > > Thank you again for your detailed feedback and suggestions. Based on your comments and input from other reviewers, we have made significant improvements to the paper’s presentation and writing to clarify our contributions and impact. Specifically, we expanded the related work section, provided additional qualitative analyses in our comment above, and refined the description of our proposed tasks to better highlight their significance.
> > >
> > > As the discussion period is closing soon, we kindly ask for your feedback on our revisions. We hope these changes address your concerns and encourage a re-evaluation of the paper’s strengths and contributions.
> > >
> > > Best regards,
> > > PARSE-Ego4D authors

---

### Official Review · Reviewer_f8JV · 2024-11-06

**Soundness:** 2
**Presentation:** 3
**Contribution:** 2
**Rating:** 5
**Confidence:** 2

**Summary:**

The paper augments Ego4D for action suggestion annotations, termed as Parse-Ego4D. The action suggestion classes include categories like search, assistant search, assistant local, language, directions, assistant guide and others. With this new proposed dataset, the paper also introduces two tasks: action recommendation systems for (1) explicit user query to action suggestion, and (2) implicit user query to action suggestions.

**Strengths:**

The motivations behind creating this augmented dataset is based on guessing future usecases in AR/VR systems. They introduce two tasks with this augmented dataset created atop ego4d.  The paper includes performance and efficiency metrics.
The subjective user study is valuable.

**Weaknesses:**

The action suggestion labels have been created solely based on narrations. Any limitations in the narrations do not have opportunity in recovering.

**Questions:**

The authors already mentioned that temporal groundings are missing. And, without temporal information either for an atomic action or a sequence of actions intuitively action suggestion for real-world captured videos can not be fully utilized. Would the authors have any insights for how that can be done?

---

> ### Author Response · Authors · 2024-11-25
> **Rebuttal**
>
> We thank the reviewer for their feedback. Here are some clarifications:
>
> **Human-derived narrations**: the narrations grounding our action suggestions come from human viewers describing the videos, providing realistic and contextually relevant perspectives. While not flawless, these human observations offer a meaningful base for developing proactive action recommendations. Future work could explore additional grounding, but we propose this dataset as a valuable initial step.
>
> **Temporal grounding**: as noted in the manuscript (line 182), each narration covers ~2 seconds of video, sufficient for generating relevant action suggestions. Small timing shifts (±5 sec) are unlikely to impact the usefulness or relevance of these suggestions in a real-world setting, especially for egocentric applications.
>
> We hope this clarifies our approach and highlights the significant value and potential of our dataset. We kindly encourage the reviewer to consider this additional context when evaluating the contribution and impact of our work.

---

> > ### Author Response · Authors · 2024-11-29
> > **Follow-up**
> >
> > Dear Reviewer f8JV,
> >
> > Thank you again for your thoughtful feedback. We’ve addressed your concerns in our response, clarifying the role of human-derived narrations and the adequacy of temporal grounding for egocentric action suggestions. We have also significantly revised our manuscript based on the other reviewers' feedback.
> >
> > As the discussion period is nearing its end, we would greatly appreciate it if you could review our rebuttal and share any further thoughts. We hope our clarifications may encourage a re-evaluation of the rating to better reflect the contributions and potential of our work.
> >
> > Best regards,
> > PARSE-Ego4D authors

---

### Meta-Review · Area_Chair_4WVc · 2024-12-12

**Metareview:**

The paper proposes a dataset for action recommendation on top of the Ego4D dataset. Reviewers appreciate the problem and data, but share concerns with the method (e.g. labels created just based on narrations, not enough data annotated by multiple participants), technical contribution (e.g. focus on dataset creation), bias, and clarity. Two borderline accept and two borderline reject are not sufficient signal to justify acceptance.

**Additional Comments On Reviewer Discussion:**

Two reviewers participated and responded to the authors' rebuttal, one positively, one less so.

---

### Decision · Program_Chairs · 2025-01-22

Reject